# HMAGLOFDB v1.0 – a comprehensive and version controlled database of glacial lake outburst floods in High Mountain Asia

Finu Shrestha[1], Jakob F Steiner[1,9], Reeju Shrestha[1,2], Yathartha Dhungel[1,3], Sharad P Joshi[1], Sam Inglis[4], Arshad Ashraf[5], Sher Wali[6], Khwaja M Walizada[7], Taigang Zhang[8,10]

[1]International Centre for Integrated Mountain Development (ICIMOD), 44700 Lalitpur, Nepal
[2]Department of Environmental Science and Engineering, School of Science, Kathmandu University, 45210 Dhulikhel, Nepal
[3]DPMM, Asian Institute of Technology, 12000 Pathum Thani, Thailand
[4]The ADM Capital Foundation (ADMCF), 999077 Hong Kong SAR, China
[5]Pakistan Agricultural Research Council, 44000 Islamabad, Pakistan
[6]Aga Khan Agency for Habitat, 44000 Islamabad, Pakistan
[7]Aga Khan Agency for Habitat, Kabul, Afghanistan
[8]College of Earth and Environmental Sciences, Lanzhou University, 730000 Lanzhou, China
[9]Department of Geography and Regional Science, University of Graz, Heinrichstraße 36, 8010 Graz, Austria
[10]State Key Laboratory of Tibetan Plateau Earth System, Environment and Resources (TPESER), Institute of Tibetan Plateau Research, Chinese Academy of Sciences, 100864 Beijing, China

*Correspondence to*: Finu Shrestha (finu.shrestha@icimod.org); Jakob F Steiner (jff.steiner@gmail.com)

**Abstract.** Glacial lake outburst floods (GLOFs) have been intensely investigated in High Mountain Asia (HMA) in recent years and are the most well-known hazard associated with the cryosphere. As glaciers recede and surrounding slopes become increasingly unstable, such events are expected to increase, although current evidence for an increase in events is ambiguous. Many studies have investigated individual events and while several regional inventories exist, they either do not cover all types of GLOF or are geographically constrained. Further, downstream impacts are rarely discussed. Previous inventories have relied on academic sources and have not been combined with existing inventories of glaciers and lakes. In this study, we present the first comprehensive inventory of GLOFs in HMA, including details on the time of their occurrence, processes of lake formation and drainage involved as well as downstream impacts. We document 697 individual GLOFs that occurred between 1833 and 2022. Of these, 23% were recurring events from just three ephemeral ice-dammed lakes. In combination, the documented events resulted in 6906 fatalities of which 906 can be attributed to 24 individual GLOF events, which is three times higher than a previous assessment for the region. The integration of previous inventories of glaciers and lakes within this database will inform future assessments of potential drivers of GLOFs, allowing more robust projections to be developed. The database and future, updated versions, are traceable, version controlled and can be directly incorporated into further analysis. The database is available at https://doi.org/10.5281/zenodo.7271187 (Steiner and Shrestha, 2023) while the code including a development version is available on GitHub.

## 1 Introduction

High Mountain Asia (HMA) has the largest expanse of glacier ice (~95000 glaciers) beyond the two poles (Guillet et al., 2022) and also has a large number (~30000) of glacial lakes, covering ~2000 km$^2$ (Wang et al., 2020). Glaciers in the region have retreated and lost mass between 0.06 and 0.4 m w.e. a$^{-1}$ across all mountain ranges since the 1960s (Bhattacharya et al., 2021), this retreat often leading the formation and rapid expansion of glacial lakes (Nie et al., 2017; Shugar et al., 2020; Zhang et al., 2021a). Numerous lakes have previously resulted in glacial lake outburst floods (GLOFs) (Carrivick and Tweed, 2013; Nie et al., 2018; Song et al., 2016; Veh et al., 2019b). GLOFs have been recorded in various parts of the world for decades (Emmer et al., 2022; Veh et al., 2022), including the European Alps (Huss et al., 2007), the Andes (Iribarren Anacona et al., 2014), North America (Wilcox et al., 2014) and the Hindu Kush Himalaya (HKH) (Ives et al., 2010; Mool et al., 2001; Rounce et al., 2017). Numerous studies suggest that glacial lakes in HMA have grown in total area and number since the 1990s (Chen et al., 2021; Shugar et al., 2020; Wang et al., 2020; Zhang et al., 2015; Zheng et al., 2021a). An overall increase of 5.9% in lake number and 6.8±0.1% in area was reported between 1990 and 2015 (Zheng et al., 2021a). Other studies find an increase by 2916 and 273.65 km$^2$ between 1990 and 2018 (Wang et al., 2020) and by 3342 and 220.64 km$^2$ between 2009 and 2017 (Chen et al., 2020). Results suggest that the total area increase has been driven by the expansion of proglacial moraine-dammed lakes (Zheng et al., 2021a; Nie et al., 2013, Gardelle et al., 2011). The number of proglacial lakes in contact with glaciers increased by 31.3± 0.3% between 1990 and 2015 (Zheng et al., 2021a), and by 96.27 km$^2$ (57%) between 1990 and 2018 (Wang et al., 2020) in HMA.

Regional studies suggest that ongoing expansion of lakes (Gardelle et al., 2011; Shugar et al., 2020) is expected to create new hotspots of hazardous lakes (Furian et al., 2022; Linsbauer et al., 2015; Zhang et al., 2022a; Zheng et al., 2021a) with implications for GLOF hazards and risk (Haeberli et al., 2016). Numerous processes have been previously identified as direct or indirect triggers of GLOFs. Dynamic slope movements (ice/snow falls, rockfalls, or landslides) into a lake can rapidly displace lake water (Awal et al., 2010; Jiang et al., 2004), similarly to glacier calving creating a displacement wave that overtops the dam (Emmer and Cochachin, 2013; Westoby et al., 2014; Worni et al., 2014). Intense rainfall or ice melt leading to sudden increases in water levels also have the potential to strain dams (Allen et al., 2016; Cook et al., 2018; Worni et al., 2012). Seismic events can destabilize moraine dams, contributing to an eventual failure (Osti et al., 2011; Somos-Valenzuela et al., 2014; Westoby et al., 2014). Seepage, piping, and degradation of an ice-cored moraine can eventually lead to dam failure (Mool et al., 2001; Yamada and Sharma, 1993). Past global assessments indicated that GLOFs have caused more than 12,000 fatalities in the last century alone and caused significant damages to infrastructure and farmland (Carrivick and Tweed, 2016). With growing populations, settlements, and infrastructural development in downstream areas, the exposure of communities and structures downstream of these lakes is rising (Li et al., 2022). Timely GLOF risk reduction measures and implementation of risk reduction strategies have become increasingly crucial but remain challenging, especially in politically sensitive regions (Allen et al., 2019; Khanal et al., 2015).

Several studies have focused on the causes, mechanisms and trends of GLOFs over the past few decades (Allen et al., 2016; Dwiwedi et al., 2000; Ives, 1986; Mool et al., 2001; Nie et al., 2020; Zheng et al., 2021c), and the number of individual studies, especially in HMA, has increased sharply over recent years (Emmer et al., 2022). A large number of these studies have focused on individual events. While some have attempted to collect information on GLOFs in HMA and have also made data accessible, they are often geographically constrained (Nie et al., 2018; Zhang et al., 2021b; Zheng et al., 2021b) or focus on certain types of GLOFs (Veh et al., 2019a). A recent global study has bridged this gap (Veh et al., 2022). However, it does not cover impacts. Recent studies have noted that a large number of GLOF events were omitted from previous records as they remained unreported or were recorded in local media and had not been documented in scientific literature (Veh et al., 2022; Zheng et al., 2021b). Studies with more detailed information on individual events either do not have accessible inventories, focus on only a certain type of GLOF (Veh et al., 2019a) or do not include more than type of lake and location (Zheng et al., 2021b). Developing more comprehensive datasets is crucial (Emmer et al., 2022), and these have become increasingly necessary as research has shifted from an evaluation of GLOF hazards to risks, including transboundary dimensions (Zheng et al., 2021a). As information on individual events may be added in future (e.g., on triggers or direct and indirect impacts) and continuously more GLOFs are reported, a database where changes can be traced and new releases are possible in the future (Blischak et al., 2016) is crucial and has been successfully employed for cryosphere data (Welty et al., 2020). Recent studies of the cryosphere have also made immense progress in demonstrating the potential benefits of traceable datasets to the scientific community and establishing standards that need to be followed to make records accessible (Mankoff et al., 2021; Welty et al., 2020). Additionally, a database with a clear structure should also be accessible by non-academic stakeholders who are not well versed in machine-readable data as well as scientists who may want to couple it to other regional datasets.

In this study, we therefore attempt to (a) provide the first comprehensive dataset of GLOFs in HMA including their location, timing, associated processes, and downstream impacts; (b) complement records from scientific literature with a rigorous evaluation of local sources on previously unrecorded events; and (c) show the potential of making such a dataset fully accessible and interoperable to couple it to other geospatial datasets.

## 2 Methods and data

### 2.1 Compilation of GLOF data

Historical GLOF data were compiled by searching peer-reviewed articles, news articles, book chapters, technical reports, and personal communication with the final cut-off date 30th June 2022. The database covers entries from 115 publications, including 83 peer-reviewed journal articles, 16 book chapters, and 16 technical reports. Additionally, online news articles (9), and social media posts (3) were also assessed. Beyond this, events reported by anecdotal accounts from local sources were also included. These were collected during fieldwork in the respective affected localities by the authors.

Correctly identifying historical events is challenging. It is highly likely that previous estimates of GLOF numbers have underestimated actual occurrences, due to a lack of accessible reports as well as many GLOFs that occurred in remote areas

with no discernible impacts on livelihoods or infrastructure (Veh et al., 2022; Zheng et al., 2021b). In this study, we have included events previously reported in scientific literature as well as reports from regional media or records from local civil society organisations and citizens. This expanded list of sources allows to reduce inherent uncertainties in each single one of them. Reported cases from scientific literature may be biased towards research catchments where scientists have a good grasp of historic cases, but these cases provide good evidence for individual GLOF characteristics. For GLOFs in HMA this is the case for a few well studied examples like Merzbacher Lake (Kingslake and Ng, 2013) or Kyagar (Round et al., 2017). News reports tend to focus on GLOFs with considerable effects on critical infrastructure or human life, and in these cases provide valuable data on impacts. Satellite imagery covers events irrespective of their impact or research interest but provides limited evidence for events that happened before the end of the 20th century. Local knowledge provides insights any of the above sources may have missed but information may have become less accurate as memory fades or while being passed on between generations. Beyond the added number of recorded events, reliance on this expanded source list introduces an inherent risk of misidentification, which must be accounted for. When debris flows or even pluvial floods reach mountain settlements they are often identified as GLOFs, without proof of their source. This happened, for example, after the Chamoli rockslide (Shugar et al., 2021) and regularly happens when debris flows occur in the Upper Indus Basin. Verification of the source is, therefore, essential. If satellite imagery is available, we verified whether the typical v-shaped moraine breach as well as deposits are visible (Zheng et al., 2021b). If available, we checked rapid lake area changes, or exposed lake beds that were clearly visible. For historic events, we ascertained whether a GLOF is technically possible (i.e., if there are lakes in the upstream). High resolution 15 cm HD Maxar satellite images (available through ArcMap) were consulted to confirm doubtful events. For events that were previously not reported and for which dates are uncertain (e.g. if only visible from satellite imagery), we also provide the date and unique identifier of the earliest satellite imagery that holds evidence of the GLOF. Being able to rely on a variety of independent sources and check with stakeholders involved in hazard response in affected regions, we are able to corroborate between multiple sources. As a result, we discarded numerous previous events as unlikely or not verifiable in a separate table (/Database/GLOFs/HMAGLOFDB_removed.csv) and did not include new events where we could not confirm local reports with satellite imagery or records from field visits.

We also record GLOFs that cannot be directly associated to a glacier, either because from the source or satellite imagery it is not clear which glacier upstream feeds into the lake or because there is no adjacent glacier in any of the available inventories. A number of regional delineations exist for HMA, including the one following RGI (Pfeffer et al., 2014), the HiMAP report (Bolch et al., 2019), an outline favoured by scientists focusing on the Tibetan Plateau (Nie et al., 2017) and the outline of the HKH by ICIMOD. None of these agree with each other, making comparisons of regional statistics very difficult. All our GLOFs are consequently georeferenced and the respective area codes for the RGI and HiMAP inventory are provided to allow aggregation. Discussing regional patterns is crucial when communicating data to policy makers, who may only feel responsible for a certain area within HMA. In the manuscript, we follow the HiMAP delineation, but bin together all subregions of the Tien Shan (inventory OBJECTIDs 1-4) to 'Tien Shan', Pamir and Alay (5 – 7) to 'Pamir', West (10) and Central (11) Himalaya to 'Himalaya West/Central', the Western Kunlun Shan (13) and the Karakoram (9) to 'Karakoram', Nyainqentanglha (18),

Gangdise Shan (17), Hengduan Shan (20) and Eastern Himalaya (12) to 'Himalaya East/Hengduan Shan' and all remaining
subregions on the Tibetan Plateau and its fringes (14 – 16, 19, 21, 22) to 'Tibet'.

## 2.2 Data structure and recorded variables

The database provides access to datafiles as well as code used to process data. All recorded GLOF events are stored in a single
*.csv file (/Database/GLOFs/HMAGLOFDB.csv), events that are doubtful or definitely not a GLOF are stored separately
(/Database/GLOFs/HMAGLOFDB_removed.csv). A separate metadata file in machine and human readable YAML (.yml)
format is provided as HMAGLOFDB_Metadata.yml, providing details on all data files and variables in the database. An
overview over all recorded database variables is provided in Table 1. Each event has a unique numerical identifier, starting at
1. Where available, the date of the event is provided up to the day of the event. Many GLOFs result in high flows or in some
cases even repeated drainage over a number of days. In this case only the last day or day of peak flood is reported, whichever
can be ascertained. When the exact year of the event is not known, we provide a range or a latest possible point in time based
on what the original source reported (*Year_approx*) or when it was first visible on satellite imagery. For events we present here
for the first time, but where an exact year is not recorded, we provide imagery IDs for Landsat images, that allowed us to
constrain the time of occurrence (*Sat_evidence*). Local names of lakes and glaciers are provided if available, should however
be used with caution for definite identification as often multiple names (or spellings) exist and sometimes different lakes are
called by the same name. Where available, the coordinates of the GLOF's source as well as the final impact location
downstream are provided as these coordinates provide the potential to prepare hazard zonation map and allow for studies
investigating actual reach of events, their impacts on downstream infrastructure, livelihoods, or ecosystems to analyse and
evaluate associated risks. While coordinates for the source are relatively easy to establish and generally somewhere within the
outline of a present or past lake, the final impact location is more difficult. We provide coordinates for the most downstream
location where the flood was observed due to high flow or any recorded local impact (indicated in *Impact_type* as *Observation*)
or, if such information is not given, the furthest location where any deposits are visible from Maxar satellite imagery (*Deposit*).
For the latter case it is hence likely that the flood reach estimate is conservative and has reached much further during the actual
event. Elevation data is taken from the original source for the event and if that is not provided, extracted from the Shuttle Radar
Topography Mission (SRTM) DEM, via the known coordinates. The type of lake (*Lake_type*), namely whether it is dammed
by a moraine or ice, or situated on bedrock, on top of (*supraglacial*) or inside a glacier (*water pocket*), is recorded, allowing
for a distinction of patterns according to the source. We also record whether the GLOF is potentially of transboundary nature
(*Transboundary*), i.e. whether the potential flow path would eventually cross a national border. It does not allow for a statement
whether the GLOF has actually crossed the border, and irrespective of date of the event assumes national borders from 2022
as given. This information is of importance as transboundary climate risks have received increased attention also for GLOFs
and other mass flows (Allen et al., 2022; Steiner et al., 2023). Location data including the country, province, river basin as
well as mountain region the source of the GLOF is located in are inherently also provided via the coordinates, however
providing this data here allows for a fast aggregation of relevant data for stakeholders not familiar with GIS data.
Information related to the GLOF event, the initial driver that caused the lake to form (*Driver_lake*) as well as the GLOF to
occur (*Driver_GLOF*) and what mechanism was at play during the drainage event (*Mechanism*) are important insights for both
regional assessments of climate drivers of these events as well as indications for dedicated numerical modelling studies
investigating drainage events. They are generally subject to great uncertainty or are simply not known at all (in both cases we
refer to the variable as *unknown*), and often only known when the site of the lake was visited right after an event. Information
on lake area (*Area*), volume drained during the flood (*Volume*), as well as flood discharge (*Discharge_water* and
*Discharge_solid*) are provided from the original sources. This information is crucial for planning, designing and
implementation of large-scale projects like hydroelectric power plants and other types of infrastructure, in order to ensure
sustainable development. These values have been obtained sometimes by measurement, sometimes by rough estimate and
uncertainties likely vary across sites, and are only rarely quantified (Kingslake and Ng, 2013; Veh et al., 2023). We finally
report a variety of information on impacts on livelihoods and infrastructure aimed to conceptualise flood-induced coping
mechanisms, enhance livelihood security, and foster self-reliance toward economic stability. This information is presented in
quantitative as well as qualitative formats enabling the database to be read by machines for regional assessments, while
retaining information that may not be readily quantifiable. Information on fatalities is categorized by gender and disabilities,
as disasters impact women and those with disabilities differently than men (c.f. Zaidi and Fordham, 2021), and the Sendai
framework on disaster risk reduction explicitly mentions the necessity of collecting "disaggregated data, including by sex, age
and disability, as well as on easily accessible, up-to-date, comprehensible, science-based, non-sensitive risk information,
complemented by traditional knowledge" (UNISDR, 2015). Differences in vulnerabilities have also been acknowledged in
HMA (Resurrección et al., 2019) but studies in the domain of disaster risk reduction investigating gender dimensions remain
rare (Halvorson, 2002; Thapa and Pathranarakul, 2019). These data are important for addressing gender inequality, cultural
beliefs, and socio-economic factors, as well as advocating for the integration of gender perspectives into disaster risk
management efforts. The lack of depth of data, i.e. when not more information is known than a total number of people killed,
also provides evidence for data collection gaps that need to be addressed in future assessments. Impacts are also quantified
through the total number of individual houses destroyed or damaged, any other infrastructure like roads or bridges impacted
(*Infra*), hydropower infrastructure damaged or destroyed (*Hydropower*) and agricultural land covered in deposits (*Agriculture*)
and large livestock killed (*Livestock*). For the few cases were records on total economic damage in monetary units exist they
are recorded separately (*Econ_damage*). Contrary to information on total fatalities, these data are less comprehensively
recorded and remain likely incomplete at the time of writing. They provide however a potential indication for regional loss
and damage assessments. Sources (scientific, media, oral) of all events are provided as full citations (*Ref_scientific_full*) or
links to the newspaper sources (*Ref_other*).
GLOFs that have been found in other sources but were found not to be associated to lake drainage or breach are stored in a
separate *.csv (HMAGLOFDB_removed.csv) with the same format as described above. The file has an additional column that
explains the reason for exclusion (*Removal_reason*). It also contains a certainty column (*Certainty*) that differentiates between
cases where we are certain it is not a GLOF, or cases where we cannot ascertain if it was one.

The database was developed keeping in mind interoperability with other datasets, enabling future local and regional risk
assessments, as well as investigations on how the occurrence of GLOFs can be explained on a regional, rather than an
individual level. Multiple datasets related to the cryosphere are readily accessible, including historic glacier outlines (Pfeffer,
2017; Soheb et al., 2022; Xie et al., 2023), elevation change (Brun et al., 2017; Hugonnet et al., 2021), glacial lakes (Chen et
al., 2021; Wang et al., 2020) and permafrost extents (Obu, 2021) These datasets can be easily combined with the GLOF
inventory for comprehensive analysis. Beyond the coordinates of lakes and downstream impact, that allow for a combination
with other spatial data, we also provide the GLIMS ID of the glacier (*G_ID*) feeding into the lake that resulted in a GLOF as
well as the ID of glacial lakes (*GL_ID*) as recorded in lake inventories.
Future events will be updated directly in the development version of the database on GitHub on a rolling basis, where additions
to the database will be visible as soon as they are updated. Each year when new events are reported, they will undergo detailed
quality check, including the documentation of information on impacts or processes that may become available weeks or months
after the event through fieldwork or detailed investigations. A new version of the database will be published under the same
database DOI (Digital Object Identifier) ensuring accurate and up-to-date information.
**Table 1: Variables for the GLOF database, as documented in the Metadata file. Format is either a string (*STR, with \* marking cases***
***where only inputs from a predescribed list are possible*) or an integer (*INT*). If an exact number is not available but is non-zero a '+'**
**is provided (e.g. 'multiple injured'). '*NA*' is used for unknown input. Note that the Metadata file itself provides further details on**
**the individual variables as well as a discussion on naming.**

| *Variable* | *Format* | *Description and unit* |
|---|---|---|
| GF_ID | INT | Unique identifier for each individual event, starting at 1 |
| Year_approx | STR | Year of occurrence; given approximately if GLOF has been identified from imagery with no definite account |
| Year_exact | INT | Exact year of occurrence |
| Month | INT | Month of occurrence |
| Day | INT | Day of occurrence |
| Lake_name | STR | Local name of the lake, if available |
| Glacier_name | STR | Local name of the glacier associated with the lake, if available |
| GL_ID | STR | Unique identifier for the lake |
| G_ID | STR | Unique identifier for the glacier (GLIMS format) associated to the lake |
| LakeDB_ID | INT | Identifier in which lakes database the lake has been mapped |
| Lat_lake | INT | Decimal degrees latitude of lake [°, WGS 84] |
| Lon_lake | INT | Decimal degrees longitude of lake [°, WGS 84] |
| Elev_lake | INT | Elevation of lake [m a.s.l.] |
| Lat_impact | INT | Decimal degrees latitude of lowest known impact of the GLOF [°, WGS 84] |

| Variable | Format | Description and unit |
|---|---|---|
| Lon_impact | INT | Decimal degrees longitude of lowest known impact of the GLOF [°, WGS 84] |
| Elev_impact | INT | Elevation of lowest known impact of the GLOF [m a.s.l.] |
| Impact_type | STR* | Quality of impact record; 'Observation' refers to lowest observed high flow or damages; 'Deposit' refers to lowest visible deposit of sediments (from satellite imagery). Impacts are therefore conservative, and are likely always further downstream than recorded |
| Lake_type | STR* | Type of lake (e.g., moraine dammed, ice dammed, bedrock etc.) |
| Transboundary | STR* | Denotes if impact is potentially transboundary |
| Repeat | STR* | GLOF that occurred again in past or future |
| Country | STR | Current national borders (2022) the source lake lies in |
| Region_RGI | STR | Region ID as used in the RGI 6.0 (Pfeffer, 2017) |
| Region_HiMAP | INT | Region ID as used in the HiMAP report (Bolch et al., 2019) |
| Province | STR | 1st level province name |
| River Basin | STR | River basin the lake is located in |
| Driver_lake | STR* | Driver that caused the lake to form |
| Driver_GLOF | STR* | Driver that caused the GLOF to occur |
| Mechanism | STR* | Mechanism involved in lake breach or drainage |
| Area | INT | Lake area [$m^2$] |
| Volume | INT | Volume of the drained lake water [$m^3$] |
| Discharge_water | INT | Measured or estimated water discharge downstream [$m^3s^{-1}$] |
| Discharge_solid | INT | Measured or estimated debris flow discharge downstream [$m^3s^{-1}$] |
| Impact | STR | Narrative description of downstream impacts |
| Lives_total | INT | Total lives lost |
| Lives_male | INT | Total male lives lost |
| Lives_female | INT | Total female lives lost |
| Lives_disabilities | INT | Total lives of people with disabilities lost |
| Injured_total | INT | Total injured |
| Injured_male | INT | Total male injured |
| Injured_female | INT | Total female injured |
| Injured_disabilities | INT | Total people with disabilities injured |
| Displaced_total | INT | Total people displaced |

| Variable | Format | Description and unit |
|---|---|---|
| Displaced_male | INT | Total male displaced |
| Displaced_female | INT | Total female displaced |
| Displaced_disabilities | INT | Total people with disabilities displaced |
| Livestock | INT | Livestock lost |
| Residential_destroyed | INT | Number of residential houses destroyed |
| Commercial_destroyed | INT | Number of commercial houses destroyed |
| Residential_damaged | INT | Number of residential houses damaged |
| Commercial_damaged | INT | Number of commercial houses damaged |
| Infra | STR | Other destroyed infrastructure |
| Agricultural | INT | Area of farmland destroyed [$m^2$] |
| Hydropower | INT | Installed hydropower capacity destroyed [MW] |
| Econ_damage | INT | Total economic damage [USD] |
| Sat_evidence | STR | Image IDs used as satellite imagery evidence |
| Ref_scientific | STR | Citation of scientific source |
| Ref_scientific_full | STR | Full references to the scientific sources |
| Ref_other | STR | Description of other sources |
| Remarks | STR | Any other remarks |
| Removal_reason | STR | Reason for removal (only for separate database of removed events) |
| Certainty | INT* | Certainty that the removed case is either definitely not a GLOF (0) or may be but was removed due to lack of evidence (1, only for separate database of removed events) |

## 3 Results

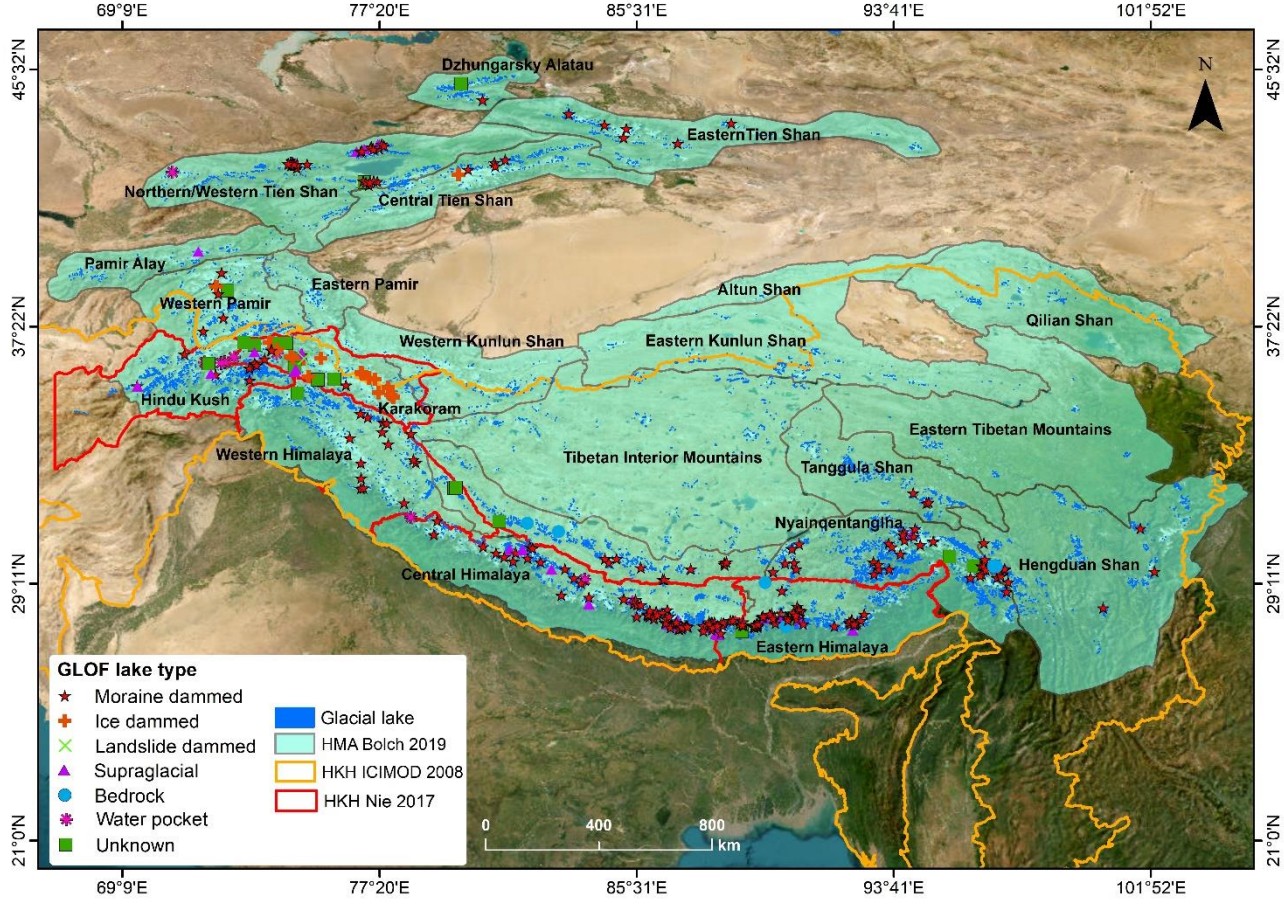

Figure 1: Overview map showing all recorded GLOFs in HMA according to the lake type. The 2018 lake inventory shown here is from ( Wang et al., 2020). The HMA outline used is following (Bolch et al., 2019). The external lake database (WangDB, ChenDB) falls within this outline. The HKH outline is based on ICIMOD (https://rds.icimod.org/Home/DataDetail?metadataId=3924) and (Nie et al., 2017)

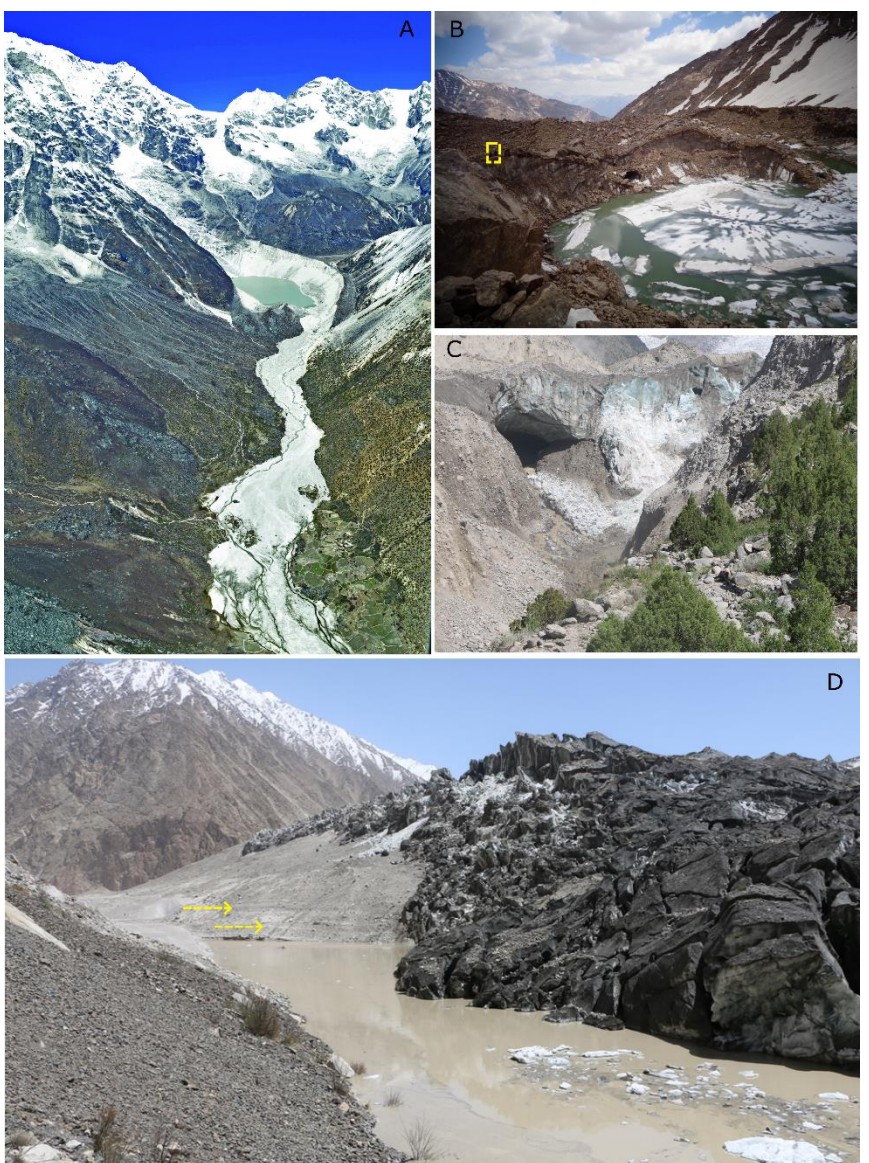

Figure 2: (A) Dig Tsho GLOF (1985; database GF_ID 322) in the Central Himalaya (Nepal) from a moraine-dammed lake. Note the settlements and agricultural land impacted in the lower reach of the deposit. The photo was taken in 2009 (Photo credit: Sharad Joshi). (B) Bam Tanab GLOF in the Hindu Kush (Afghanistan) in 2021 from a supraglacial lake (GF_ID 510). The ice tunnel, through which the lake drained is visible in the center. Note the two people sitting on top of the ice cliff (yellow square). The photo was taken several days after the event (Photo credit: Milad Dildar). (C) Ice tunnel exit at the terminus of the tributary to Badswat Glacier that caused a GLOF in 2018, eroding substantial amounts of the moraine in the immediate downstream (Karakoram, Pakistan, GF_ID 493; Photo credit: Sher Wali). (D) Ice dam at Khurdopin Glacier caused by a surge. The terminus moraine is visible in light grey on the left, partly covered by the spillover ice from the surge. The lake at this location refills multiple times after the surge, sometimes up to half of the moraine height (note the water line is slightly visible from a change of colour, yellow arrows). Photo taken from the location where the lake drains under the surged ice (multiple events starting from GF_ID 26; Photo credit: Sher Wali).

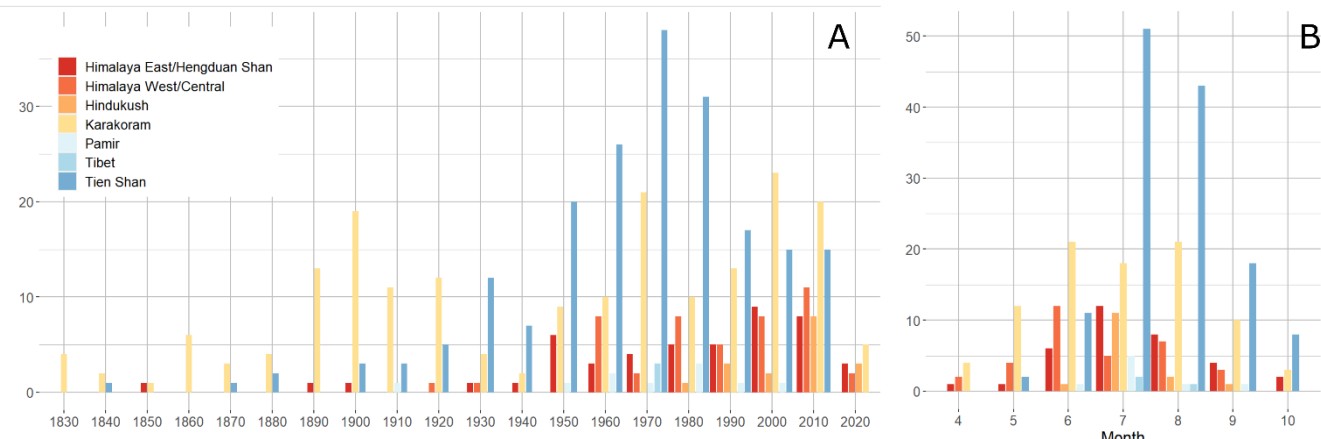

**Figure 3: (A) Temporal evolution of recorded GLOF occurrence per region and decade. (B) Seasonal occurrence of GLOFs. Months November to March are not shown, with total events <5.**

Our database includes 697 individual GLOF events (Figure 1, Figure 2), recorded between 1833 (with 4 historic events before that date, where any validation of processes is virtually impossible) and 2022 (Figure 3). 7% (46 of 697) of the reported GLOFs here have not been documented in any previous study. Conversely, 101 previously reported GLOFs were removed based on new evidence to the contrary (17) or a lack of strong evidence (84) for it being a GLOF.

28.4% of the events in the database were recorded in China, 24.7% in Kyrgyzstan, 21% in Pakistan, 8.5% in India, 7.6% in Nepal, 4.9% in Kazakhstan, 2.9% in Bhutan, 1.6% in Tajikistan and 0.6% in Afghanistan. Following the RGI delineation, 30% of the GLOFs were recorded in the Karakoram, 18% in the Eastern Himalaya and 29% in the Western Tien Shan. Due to different choices in delineations, this looks considerably different for the HiMAP outlines where 28% are in the Karakoram, 17% in the Northern/Western Tien Shan, 14% in the Central Himalaya and 10% in the Eastern Himalaya, outlining the sensitivity of statements regarding mountain ranges to the choice of delineation product.

For 325 events (47%) we know the month of occurrence and, of these, 74% took place between June and August (Figure 3B). Only 3% of all GLOFs were recorded between November and March. For 275 events (39%) the day of the GLOF is known, potentially allowing for an analysis of prevailing weather patterns preceding the outburst.

The mean elevation of individual lakes associated to a GLOF was 4598 m a.s.l. (336; min = 2562 m a.s.l., max = 5982 m a.s.l.). For the GLOFs where impacts were recorded (459; 66%) the mean elevation difference between lake and lowest recorded impact was 1161 m a.s.l. (min = 19 m a.s.l., max = 4431 m a.s.l.).

47% of GLOFs originated from moraine-dammed lakes, 34% from ice-dammed and 10% from supraglacial lakes (Figure 1). Other types occurred infrequently, and particular glaciers were associated with repeated flood events, including englacial outbursts from water pockets either deep in the ice or superficially around crevasses (3%), outbursts from bedrock lakes in periglacial terrain (1%) and outbursts from landslide dammed lakes that formed downstream of glaciers (0.4%). Multiple events in the Bagrot subbasin in the Upper Indus, as well as in the Ala Archa basin in Kazakhstan may be drainages from water pockets, based on the nature of drainage and the lack of evidence for lakes on the glacier surface or the periglacial terrain, but

local evidence is too uncertain to confirm this. For the Ala Archa basin, we have removed many cases, as even previous authors did not insist on all being GLOFs (Medeu et al., 2016). These events all occurred at the terminus of very steep glacier tongues and some observed characteristics of these events (e.g. melt water to the base resulting in ice and terrain collapse) resemble glacier detachments (Kääb et al., 2018). However, the eventual runout as a debris flow with high water content is markedly different. For 4% of the cases, the lake type was unknown, especially for GLOFs for which no satellite imagery exists for validation.

Our knowledge of the mechanisms of how lakes are formed (known for 20% of the cases), triggering (12%) and failure mechanisms of the outburst (26%) remains limited (Figure 4). Even less is known about the drained total volume ($m^3$, 15%), discharge of water ($m^3$ $s^{-1}$, 10%) or discharge of solids ($m^3$ $s^{-1}$, 3%), important variables for modelling studies. Additionally, these reported values are often associated with large uncertainties.

Of all GLOFs, 2 were located in an area where no glacier could be found anywhere upstream. However, they are located in areas that were most likely glaciated. 22 GLOFs occurred below glaciers that were not mapped in RGI 6.0, showing the still considerable number of glaciers that are missed from this global inventory. Similarly, 10 GLOFs (at moraine-dammed lakes) were recorded where no lake was apparent in any satellite imagery. This is possible for GLOFs recorded before the satellite age. In 97 cases, a lake or depression was visible but not mapped in any of the inventories. In 196 cases (28%) the lake is ephemeral and hence not in any inventory, especially the case for ice-dammed or supraglacial lakes. Some ice-dammed lakes were in inventories if they were present at the time of the inventorization. However, when comparing these data to a GLOF, their area should be taken with caution, considering their rapid change with a change in meltwater availability.

25 GLOFs (4%) have 6906 recorded fatalities (Table 2), although 6000 were attributable to a single event in India in 2013, where fatalities were caused by a multitude of factors in a complex compound event (Allen et al., 2016). This latter event, until recently, accounted for nearly 50% of recorded global GLOF deaths (Carrivick and Tweed, 2016). Nearly all casualties have been attributable to GLOFs from moraine-dammed or supraglacial lakes, while no fatalities have been linked to ice-dammed lake breaches (Table 2), while there may of course have been unrecorded impacts (Hewitt and Liu, 2010). Numbers on injured or displaced remain rare, but are likely much higher, especially considering people moving away, many in the months after an event due to the long-term negative impacts of damaged infrastructure. Nearly 2000 livestock were reported killed. More than 2200 residential and commercial buildings were reported destroyed or significantly damaged and numerous bridges were destroyed. At least 71 $km^2$ of agricultural land was destroyed and hydropower structures with a combined capacity of 164 MW were destroyed or heavily damaged. In only a few cases (13) were estimates of economic damages in monetary values attempted (5.3 billion USD). These were limited to damages associated with the flooding and did not include considerations of the longer-term economic toll of damaged infrastructure, disablement, destroyed farmlands, or the long-term impacts on accessibility of health, education, or market facilities due to impacts on transport infrastructure.

**Table 2: Number of GLOFs per region and type of lake with associated fatalities (N/fatalities).**

| Lake type/Region | Pamir | Tien Shan | Tibet | Hindu Kush | Karakoram | Himalaya West/Central | Himalaya East/Hengduan Shan |
|---|---|---|---|---|---|---|---|
| Moraine-dammed | 6/25 | 96/65 | 4/0 | 10/20 | 7/0 | 78/6236 | 129/454 |
| Ice-dammed | 4/0 | 86/0 | 0/0 | 0/0 | 144/0 | 0/0 | 0/0 |
| Supraglacial | 1/100 | 17/0 | /0 | 3/0 | 26/1 | 19/0 | 6/0 |
| Others/unknown | 1/0 | 15/0 | 0/0 | 6/3 | 19/2 | 1/0 | 19/0 |
| TOTAL | 12/125 | 214/65 | 4/0 | 19/23 | 196/3 | 98/6236 | 154/454 |

338 lakes were associated with 697 GLOFs, with 61 (18%) of the lakes causing a GLOF more than once and the 17 (5%) lakes releasing GLOFs at least five times, made up 43% of all GLOFs (Table 3). The return period of some of these GLOFs is rapid, occurring repeatedly over consecutive years such as at Merzbacher or Khurdopin, while for others a GLOF may occur more than decades apart. In many cases, for ice-dammed lakes, this is coupled with the return period of the respective glacier surge. Many of these lakes have resulted in GLOFs until very recently and we hence consider *active*. Others, like the GLOFs at Chong Kumden (ice-dammed), are considered inactive as the tongue has receded so far that even during a surge it cannot block the valley anymore. For Salyk or Topkaragay (moraine-dammed), the tongue recession has presumably also resulted in a lack of direct meltwater supply to the local depression. While the three lakes draining most frequently (and all still *active*) are all ice-dammed, repeat GLOFs are common from all major types of glacial lakes (Table 3). However, repeat drainages are decidedly uncommon in the Himalaya or the Tibetan Plateau.

190 (27%) were potentially transboundary GLOFs, i.e. their flood could have crossed a border further downstream, 55 of which originated in China. Fewer than 10 of these GLOFs have however actually recorded impacts across borders (China to Nepal and Uzbekistan to Kyrgyzstan).

**Table 3: Lakes with more or equal to five recurring GLOFs in HMA.**

| Lake/Glacier name | Lat (º) | Lon (º) | Elev (m a.s.l.) | Region | Outburst recurrence | Period of GLOFs | Lake type |
|---|---|---|---|---|---|---|---|
| Merzbacher/ Southern Inylshek | 42.20 | 79.85 | 3271 | Central Tien Shan | 86 | 1902 - 2015 | Ice-dammed |
| Khurdopin/Khurdopin | 36.34 | 75.47 | 3482 | Karakoram | 37 | 1882 - 2021 | Ice-dammed |
| Kyagar/Kyagar | 35.68 | 77.19 | 4880 | Karakoram | 34 | 1880 -2019 | Ice-dammed |
| Unnamed/Aksay | 42.53 | 74.54 | 3637 | Northern/Western Tien Shan | 30 | 1877 - 2015 | Moraine-dammed |

| Lake/Glacier name | Lat (º) | Lon (º) | Elev (m a.s.l.) | Region | Outburst recurrence | Period of GLOFs | Lake type |
|---|---|---|---|---|---|---|---|
| Unnamed/Kuturgansuu | 42.52 | 74.61 | 3470 | Northern/Western Tien Shan | 17 | 1846 - 2010 | Moraine-dammed |
| Unnamed/Chong Kumden | 35.17 | 77.70 | 4691 | Karakoram | 14 | 1533 - 1934 | Ice-dammed |
| Hassanabad/Shisper (both names for lake and glacier) | 36.39 | 74.51 | 3370 | Karakoram | 13 | 1894 - 2022 | Ice-dammed |
| Karambar/Karambar | 36.62 | 74.08 | 2935 | Karakoram | 11 | 1844 - 1994 | Ice-dammed |
| Unknown/Teztor | 42.54 | 74.43 | 3606 | Northern/Western Tien Shan | 11 | 1910 - 2012 | Moraine-dammed |
| Ghulkin/Ghulkin | 36.42 | 74.88 | 2692 | Karakoram | 8 | 1980 - 2009 | Supraglacial |
| Lake number 6/ Glacier No 182/Bezymyannyi/TEU-Severny | 43.14 | 77.28 | 3380 | Northern/Western Tien Shan | 8 | 1973 - 2014 | Supraglacial |
| Unnamed/Halji | 30.27 | 81.48 | 5347 | Central Himalaya | 6 | 2004 - 2011 | Supraglacial |
| Unnamed/Salyk | 42.52 | 74.72 | 3390 | Northern/Western Tien Shan | 6 | 1938 - 1980 | Moraine-dammed |
| Unnamed/Topkaragay | 42.49 | 74.52 | 3680 | Northern/Western Tien Shan | 6 | 1928 - 1993 | Moraine-dammed |
| Unnamed/Central Rimo | 35.42 | 77.61 | 5100 | Karakoram | 5 | 1976 - 2014 | Ice-dammed |
| Unnamed/Batura | 36.51 | 74.85 | 2713 | Karakoram | 5 | 1873 - 1974 | Supraglacial |
| Unnamed/North Terong | 35.25 | 77.31 | 4400 | Karakoram | 5 | 1975 - 2002 | Ice-dammed |

308

309

310

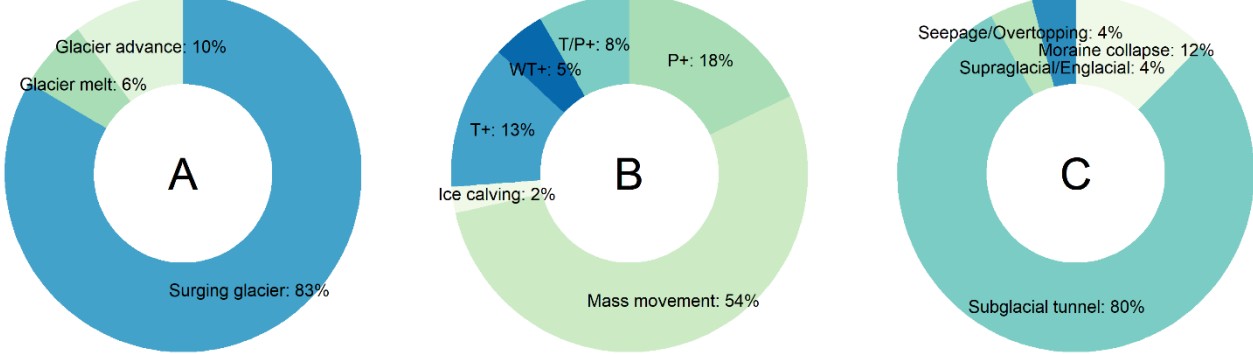

311

**Figure 4: Different drivers that caused the lake to form (139) (A), caused GLOF to occur (84) (B), and mechanisms involved in lake breach/drainage (178) (C). 'Glacier melt' refers simply to melt water provision from any adjacent glacier, irrespective of their state of retreat, advance or stability. T, P and WT+ stand for reported temperature, precipitation and water table and temperature increases, respectively. Mass movements include ice and rock avalanches as well as landslides, debris flows and other flood events. Moraine collapse is most often characterized by ice core thawing. 'Seepage/Overtopping' also includes piping through the dam.**

## 3.1 Uncertainty of individual variables

GLOFs often occur in remote areas, where reports are scant and confirming process chains and quantifying associated volumes and impacts is challenging. Here we discuss the uncertainties related to variables documented in this database. For many GLOFs (181; 26%) even the year of occurrence is uncertain, which is reflected in the database by either providing NA or a time window during which it must have happened. Days of occurrence (available for 39% of the records) are often also different between reports on the same event, either due to erroneous records or the magnitude of certain GLOFs resulting in floodwaters reaching areas over more than one day. Any recorded day hence likely has an uncertainty of +/-3 days. Coordinates of the source lake are correct if provided, however, do not reflect the exact location of the breach but any coordinate within the perimeter of the lake. Coordinates of the impact area are a lot more uncertain. If provided they show either the lowest location where deposits are still visible on Maxar imagery from 2021 (*Deposit*) or where high flow or damages were reported (*Observation*). Naturally, this is a conservative estimate and high flows have likely reached thousands of meters further downstream and with deposition or erosion occurring along the riverbed that may not be visible on imagery. Elevation values are generally retrieved via the coordinates from the Shuttle Radar Topography Mission (SRTM). Considering the steepness of the terrain across HMA, we expect an unknown uncertainty of the actual elevation of the source, and values should be considered an indication of the elevation range, rather than for example an exact input for a hydrodynamical model.

For a few events, the primary source is local informants, while for some events that have been reported in news articles or by other studies, local sources serve as a way of confirming the event and the associated recorded variables. While local sources have their own uncertainties, including misidentification of another mass flow event as a GLOF or misrepresentation of impacts to either gain attention for compensation (and hence exaggerating impacts) or dispelling concerns of potential tourists (hence underestimating impacts), they also can provide insights that other sources cannot. When the source lake is regularly visited due to agricultural activities or for leisure, locals are often able to make statements on what likely outburst mechanisms were or what caused the event in the first place. Observations on the discharge hydrograph, especially whether the discharge peak arrived rapidly or gradually, provide indications on the type of drainage mechanism.

When reporting areas of lakes associated with GLOFs, we refer to the reported value provided in the respective studies. For the few events we report here for the first time and if satellite imagery is available a few days before the event, we make estimates based on that imagery. Lakes linked to GLOFs often exhibit rapid areal changes in the weeks and days prior to eventual failure of the dam or opening of a tunnel. This is especially true for ephemeral ice-dammed lakes (Muhammad et al., 2021; Round et al., 2017; Steiner et al., 2018). Therefore, the lake area mapped many weeks or even months before a drainage event is not necessarily a good proxy for the actual water volume that drained. We report estimated drained volumes and sometimes even estimates of discharge exist. However, the accuracy of such estimates is impossible to verify and

measurements during GLOFs are rarely possible (Muhammad et al., 2021), and rating curves in fast changing river beds are
unlikely to be accurate. Such data should hence be considered as first order indications of magnitude rather than accurate
representations of actual discharge.
Estimates on impacts are conservative. Based on our observations from fieldwork and the concurrent reporting of hazards,
media coverage often overlooks to report on remote villages and numbers of mortalities or valuations of infrastructure impacted
are prone to deflation or inflation as a result of inaccurate transmission of information or wilful tampering with data. This can
happen for various reasons, as evidenced through fieldwork across the region. The government not reporting impacts or making
them look smaller than they are can be favourable as this would otherwise provide arguments for becoming more active in
mitigation response, which they may not be able or willing to do. Reports on high impacts are often also perceived as damaging
to tourism and hence discouraged. In light of financial reparations, either from the national to the local governments or
communities or by the global community to a country, overreporting of impacts is naturally preferred. If impacts in doubtful
cases could not be verified by additional sources, they were not reported. Reported data from scientific literature were
dismissed in rare cases where for example a GLOF was reported twice within two days, and it was likely that this happened
due to erroneous reporting dates.
**4 Discussion**
In the section below, we discuss the potential of the GLOF database for comparisons with existing and future data sources
related to regional assessments and investigations of individual events. We then conclude by comparing the database to
previous efforts and address its inherent limitations.
**4.1 Temporal and spatial trends**
The discussion on whether GLOFs are increasing with a changing climate trend has been dealt with elsewhere in detail
(Harrison et al., 2018; Veh et al., 2022). Here, we only provide a brief discussion that stems from the compilation of the data,
and that is crucial when using databases of hazard events in remote areas like mountains. As we rely on different source types,
namely previous academic publications, news and technical reports, satellite imagery and local knowledge from involvement
in fieldwork in numerous locations across HMA, we are able to complement the deficiency any single of these approaches
may have and provide some perspectives on trends that may be due to observational bias. Figure 3A shows a clear and rapid
increase of recorded events in the Tien Shan from the mid 20th century with a subsequent decrease in the 1990s. There is little
indication for this being climate-related and we hypothesize this is a function of increased visibility thanks to the extensive
efforts of scientists at the end of the Soviet era to monitor debris flows (and as a consequence related GLOFs), which died
down as the Central Asian states became independent. This aligns exactly with the development and subsequent demise of
cryosphere monitoring in the area between 1950 and 1990 (Hoelzle et al., 2019). Medeu et al. (2019) on the other hand argue,
that at least for Kazakhstan a decrease in debris flows from glacial sources can be attributed to the successful monitoring and
lowering of lakes in the region since the late 1990s. Conversely, for the Karakoram, there are many peaks in observed events
starting at the turn of the 19[th] century. This may be explained either by surge cycles occurring around the same time, resulting
in ephemeral ice-dammed lakes, or the piqued interest of the British Empire around the turn of the century in the region during
the Great Game (most reports on GLOFs stem from British officers stationed in the region, cf. Hewitt and Liu, 2010). Interest
decreased until Independence and finally grew again towards the end of the century as seen in other areas with increased
infrastructure development, starting with the construction of the Karakoram Highway between the 1960s and 1970s
(Kreutzmann, 1991) and the arrival of media in remote areas. There is a relatively consistent increase of reported events in the
Hindu Kush and the Himalaya region, albeit total events are much lower (i.e. <1 event yr$^{-1}$ for a whole region). We believe
that our records of events have also increased due to an increase in interest in the topic in the region (Emmer et al., 2022),
which led to more detailed documentation of individual events by various scientists.
The seasonal patterns (Figure 3B) are instructive as they show a clear earlier peak in the year for the Himalaya region, a more
stretched out peak in the Karakoram and shift in the Tien Shan, influenced by Westerlies rather than the South Asian Monsoon,
providing an indication for the potential of using such a dataset to investigate the importance of regional precipitation patterns
for drivers of outburst floods.
The database records four GLOF events in Afghanistan, three of which for the first time in this database, with 15 fatalities and
more than 300 houses destroyed, and one from a previous publication only based on geomorphic evidence with no recorded
impacts. This puts the country on the map with respect to GLOF hazards and emphasizes the importance of local data for
accurate assessments, considering that these events occurring between 2013 and 2021 remained previously unnoticed in
literature. Due to the lakes located on a debris-covered surface draining englacially, hence leaving no clear evidence of a dam
breach, potentially visible from satellite imagery, they also constitute examples that remain difficult to identify with automated
approaches relying on satellite imagery only.

## 4.2 Comparisons to other regional data

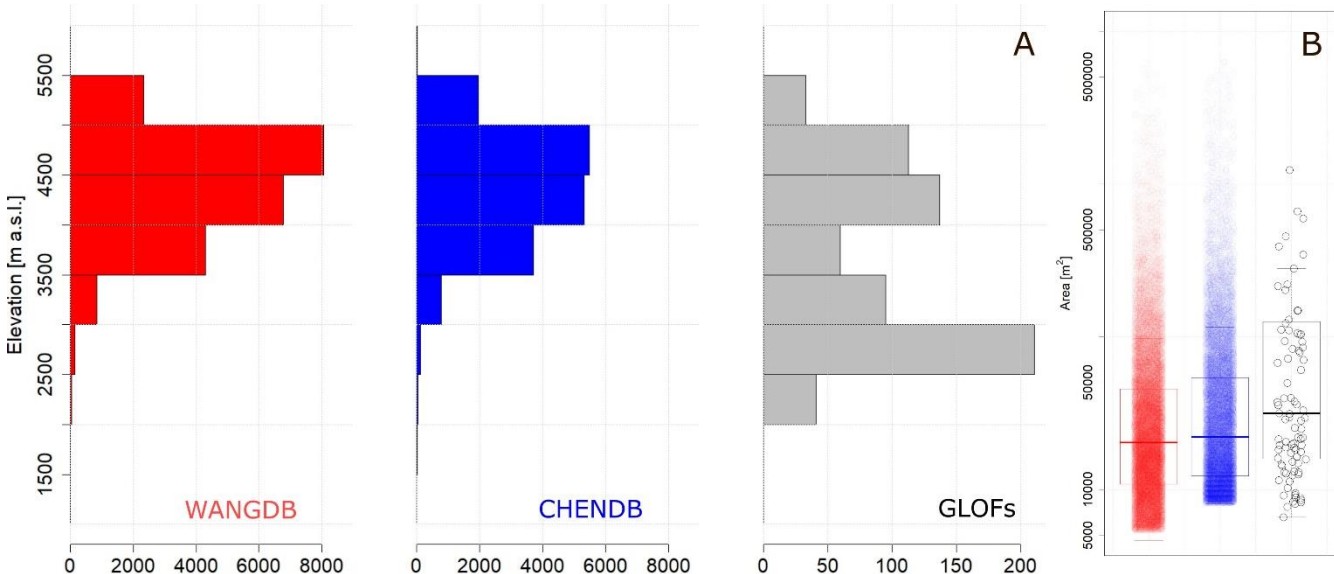

Figure 5: (A) Elevational distribution of unique lakes recorded in databases, covering two years 1990 and 2018 (WANGDB, Wang et al., 2020) and ten years between 2008 and 2017 (CHENDB, Chen et al., 2021), compared to only those lakes that resulted in GLOFs (GLOFs). (B) Lake areas of unique lakes in the same databases as well as specifically for the lakes that resulted in GLOFs, where available.

Combining the lake and GLOF inventories allows for a comparison of topographic as well as meteorological conditions which may have contributed to the occurrence of GLOFs and those that did not. This is useful for the identification of potentially dangerous glacial lakes (Ahmed et al., 2022; Ashraf et al., 2021, 2012; Bajracharya et al., 2020; Bolch et al., 2008; Duan et al., 2020; Zhang et al., 2022b). The evaluation of meteorological conditions would require an in depth discussion of the respective datasets and wider synoptic conditions, attempted previously for mudflows (Mamadjanova et al., 2018). The database suggests that mass movements in the lake's vicinity, as well as intense precipitation immediately preceding the GLOF are important drivers (Figure 4B), but only 12% of all entries record any driver and they are generally only based on local assessments that may not provide the definite reason. As quality controlled reanalysis data becomes more readily available at the kilometre scale for HMA (Wang et al., 2021), more comprehensive analysis of climate drivers become possible and would elucidate, which climate variables are crucial to understand with respect to a changing climate, to anticipate changes in GLOF occurrence.

A comparison to already established inventories of glacial lakes is more straightforward. Figure 5 shows elevational and area statistics of lakes from two different inventories (Chen et al., 2021; Wang et al., 2020). While both inventories rely on Landsat data for identification, one follows a manual approach in identifying the outlines in 1990 and 2018 (Wang et al., 2020; henceforth referred to as WANGDB), while the second delineates automatically at the 30 m pixel resolution (Chen et al., 2021; CHENDB), allowing for decadal data between 2008 and 2017 but a coarser resolution of outlines. Both inventories include

lakes from different time steps, and we consider a lake that is present in more than one time step once, at whatever earliest
year it appears. The GLOF database allows us to quickly visualize, that while lakes follow a unimodal distribution over altitude
similar for both approaches of delineation, GLOFs are bimodally distributed with peaks above 2500 and 4000 m a.s.l. (Figure
5A). This provides a clear indication of lakes at lower elevation being relatively more susceptible to an outburst, but no clear
decrease of the risk with elevation. Of the 339 lakes that resulted in GLOFs, 99 (29%) appear in inventories at one point and
we hence have an indication of their size (Figure 5B). While CHENDB covers more time steps, the minimum lake area it can
detect (8100 m$^2$) is nearly twice as large as for WANGDB (4600 m$^2$). 9 out of the 99 lakes with area data that resulted in
GLOFs, hence only appear in WANGDB. On top of the many lakes that do not appear in inventories at all (71%) due to their
ephemeral nature or being present before the coverage of inventories, the possibility that potentially dangerous glacial lakes
are completely missed by lake inventories relying on satellite imagery hence needs to be considered.
**4.4 GLOF paths**
Records of the extent of GLOF events (459) allow for an evaluation of GLOF paths (Figure 6). Exploiting other spatial datasets,
this can help in evaluating possible drivers as well as impacts. Repeat spatial products related to the cryosphere available in
HMA include decadal glacier outlines (He and Zhou, 2022; Lee et al., 2021; Xie et al., 2023), distributed data on ice mass loss
(Brun et al., 2017; Hugonnet et al., 2021), permafrost probability (Obu, 2021) and snow cover (Muhammad and Thapa, 2021).
Such datasets can be compared with recorded GLOF events and lakes that have not resulted in GLOFs to evaluate potential
drivers. The hazard of a GLOF is closely associated with moraine stability, which is linked to the immediate history of glacier
retreat in the area. Areas of recent permafrost change are likely also more susceptible to mass movements, which have already
been identified as important drivers of GLOF events (Figure 4). Rapid increase in local temperature is often recorded as a
reason for GLOFs occurring (Figure 4), but to date no studies exist that establish a definite link between increased melt from
either ice or snow that eventually results in the drainage of a lake or failure of a dam.
On the downstream, distributed datasets on infrastructure, population or ecosystems allow for assessments of impacts and
vulnerabilities (Figure 6). Coupling of GLOF paths with distributed population data (Thornton et al., 2022) would allow for
the computation of people potentially impacted, coupled with remotely sensed vegetation and agriculture data which would
allow for estimates on local economic and ecological impacts. GLOF paths could also be used to develop hazard zonation
maps, as is already standard practice for avalanches in many regions.

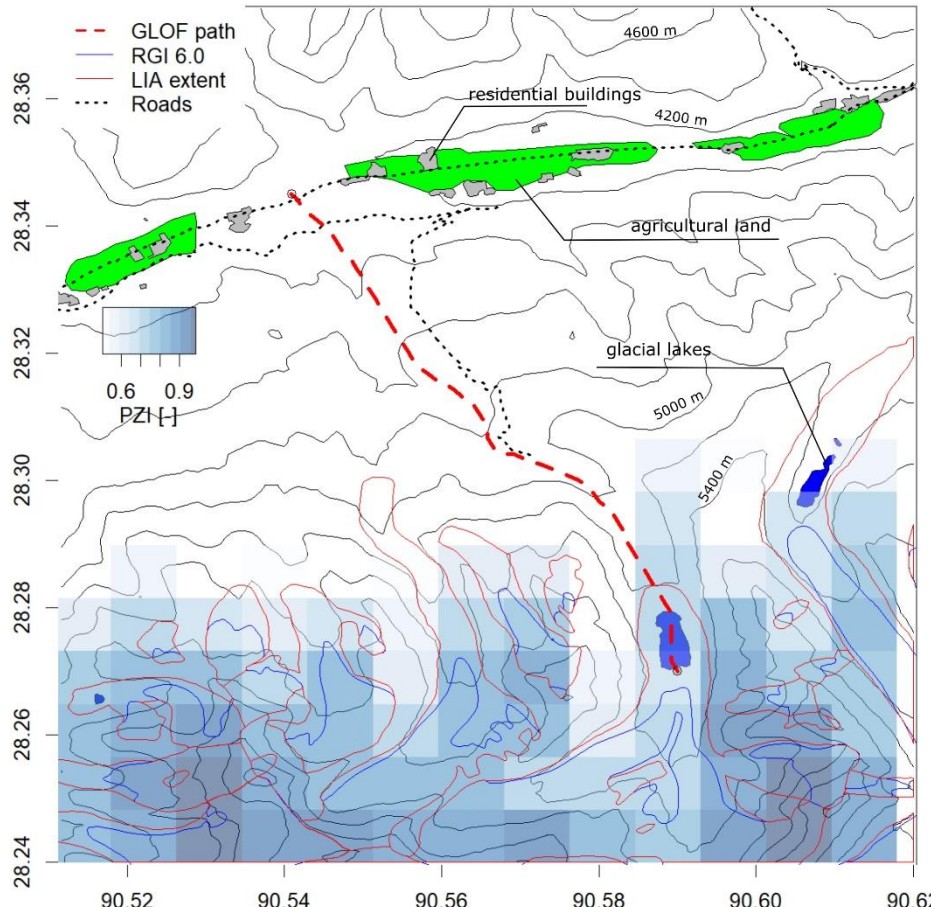


Figure 6: Extracted GLOF path for one event (GF_ID 651), showing the RGI 6.0 outline, the possible glacier extent during the little
ice age (Lee et al., 2021) and probability of permafrost occurrence, with high values suggesting a likely and low values a less likely
occurrence of permafrost (PZI, Obu, 2021). Shape files of roads are taken from Open Street Map (OSM), residential and agricultural
areas are mapped from Maxar imagery. The figure can be directly created in the available R code from the database for any event.

452

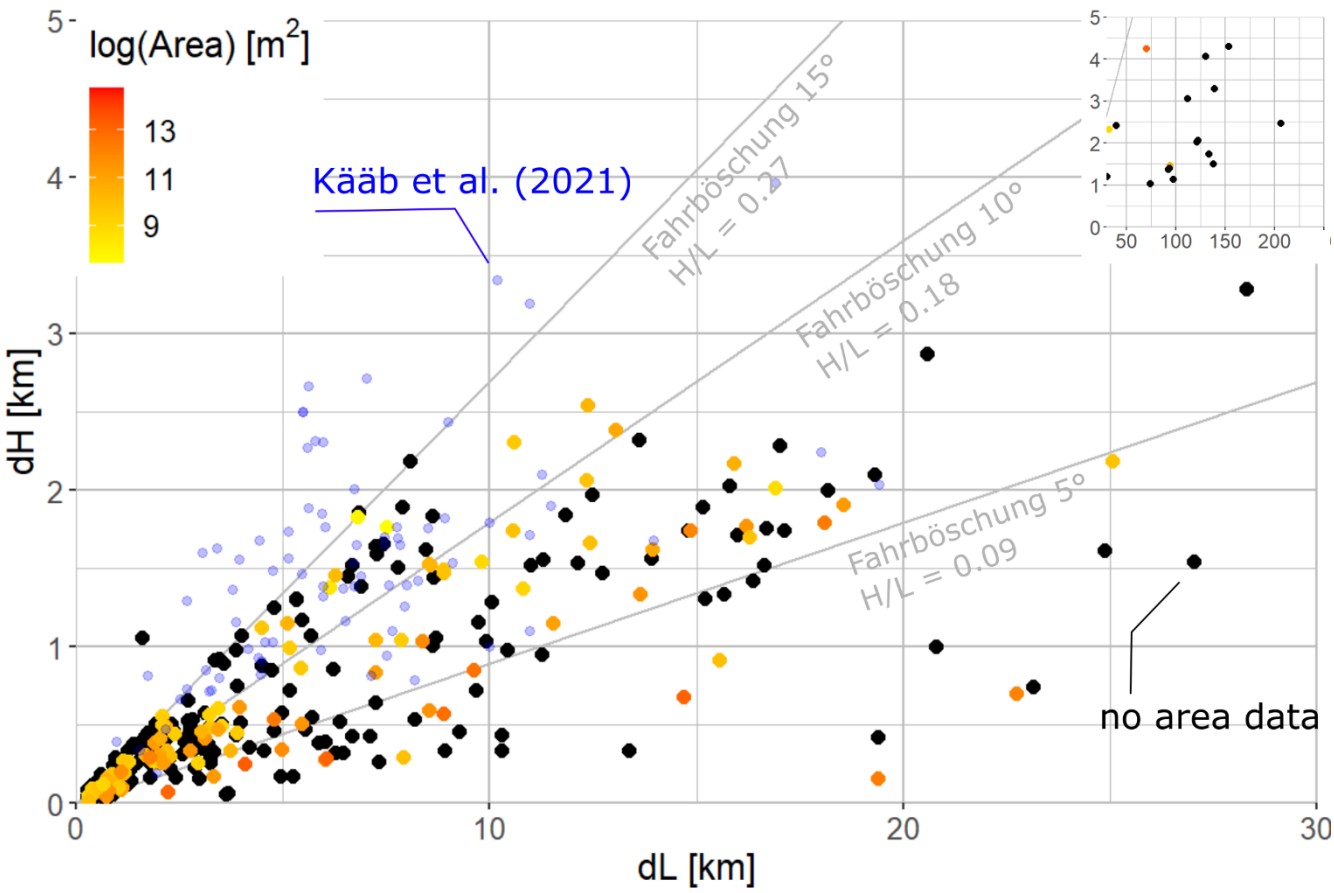

453

**Figure 7: Reach angles (*Fahrböschung*) for all GLOF events with recorded downstream impact location (459). Black dots are events where no recorded lake areas exist. Shaded from yellow to red are GLOFs for which some record of lake area exists (either from the original source or a lake database), that may not necessarily be the lake area just before the event. In light blue records from Kääb et al. (2021) are shown, which largely relate to glacier detachments or historic large scale debris flows and GLOFs.**

For GLOFs where the location of the lake and the lowest recorded evidence of the outburst flood (either reported flood or observed debris deposits), we are able to calculate reach angles, i.e. the elevation difference over the distance of the flood path between start and end point (Figure 7). Current studies estimating the future exposure of downstream infrastructure and livelihoods to potentially dangerous glacial lakes are based on estimated runouts (Schwanghart et al., 2016; Taylor et al., 2023; Zheng et al., 2021a), without the ability to rely on actual expected reach. While a definite prediction of any future GLOF reach will never be possible and even detailed numerical studies of individual events are sensitive to small changes in topography (Muhammad et al., 2021; Westoby et al., 2014), the data presented here allows future studies to apply ranges of a potentially likely reach, depending on glacier size and path topography. With a median runout length of 6.1 km (mean μ = 32 km, biased towards a few events of exceptional reach; Figure 7 inset) and a median elevation drop of 1036 m (μ = 1126 m), the mean reach angle is 0.14 (σ=0.09; 8°). Nearly all of the events markedly cluster below a reach angle of 0.27 (15°, Figure 7), considerably shallower than large scale mass flow events like glacier detachments (Kääb et al., 2021). For comparable drop

heights GLOFs hence travel considerably further than glacier detachments, but the data also suggests a limit in reach, that can
provide an upper bound when assessing the potential reach of lakes at risk of an outburst in future. While it would be possible
to establish lake volumes from lake areas (Cook and Quincey, 2015), the inherent uncertainties in these approaches and the
discrepancy between date of lake mapping and GLOF event result in large overall uncertainties trying to establish a link
between lake volumes and potential reach of GLOFs. We therefore only use areas from the available inventories or mapped
areas from just before the event as a proxy (Figure 7). Lakes with an area larger than $10^5 \, m^2$ (123) result in median reach angles
of 0.16 (9.0°), lakes smaller (107) in slightly steeper angles of 0.17 (9.8°), suggesting that the size of a GLOF holds some
potential for projecting its reach, apart from the many other influencing factors.

## 477 4.5 Downstream impacts

Excluding the single Kedarnath event in 2013, wherein 6000 people were reportedly killed by a combination of other hazards
including but not limited to the GLOF, 906 people were reported to be killed directly by 24 GLOFs. This number is three times
higher than the previous record of 300 (Carrivick and Tweed, 2016). Only considering the time until 2005 to cover the same
periods, our database still records 854 fatalities. Only 25% (54) of all recorded events (216) in the region had previously any
impact recorded (Carrivick and Tweed, 2016), while our database now has impacts on livelihoods or infrastructure noted for
149 events (21%). Comparing this number to other mountain hazards in the region is difficult, as data are rare. Snow and ice
avalanches (excluding mountaineering accidents) in the same region have resulted in more than 3000 fatalities, with records
only starting in the 1970s (Acharya et al., 2023). Landslide fatalities for countries entirely in HMA suggest 67 deaths in
Afghanistan, 50 in Bhutan, 809 in Nepal and 75 in Tajikistan between 2004 and 2010 (Petley, 2012), suggesting this hazard
to be considerably more dangerous to human life.
Impacts beyond fatalities are often recorded, but socioeconomic valuations are lacking. In the few individual events with
valuations of damage by the immediate event, this is typically focused on the value of expensive national infrastructure like
hydropower projects. The value of agricultural land, residential or commercial buildings, or long-term damage due to damage
to road, health or education infrastructure are rarely assessed. Infrastructure datasets that allow for a rapid assessment of
damage are generally not available. Such an assessment was previously attempted for GLOFs globally (Carrivick and Tweed,
2016). However, a more thorough discussion of available impact datasets for HMA is required. Many of these data are available
in reports prepared for disaster response (e.g. for the Upper Indus basin, both Afghanistan and Pakistan, prepared by AKAH
or UNDP (Ashraf et al., 2015; Gohar, 2014) or Kazakhstan prepared by scientists from local Universities (Medeu et al., 2016))
but have so far not been compiled in a standardized manner.
A few GLOF events have now been recorded already that have reached urban (e.g. GF_ID 131 south of Almaty, where dam
structures were erected as adaptation measures) and semi-urban (e.g. GF_ID 515 north of Melamchi) space. As more and more
infrastructure is being built in upstream areas and close to river channels, vulnerabilities are expected to increase. Future studies
should investigate potential flood impacts on built up areas as well as hydropower infrastructure, which often require different
model setups than for environments with infrastructure only including roads and single houses (Fischer et al., 2022).
While nearly one-third of all GLOFs could potentially be transboundary (i.e. the downstream path eventually crosses a current
national border), less than ten events have resulted in confirmed transboundary impacts. While a lot of attention has been given
to transboundary climate risks in the recent past, our data suggests that it may be less the actual flood wave that is of
transboundary concern and more the associated impacts in one country (e.g. road disruptions with impacts on trade) that could
have knock-on effects across borders, including a disruption of trade, access to health care or education (Steiner et al., 2023).
Mass flow events like GLOFs have been previously identified as an important part of the sediment balance in mountain rivers
(Cook et al., 2018), with potential implications for landscape formation as well as hazards (Li et al., 2022). The database does
include recorded values of discharge of water (72; $\mu = 3248$ m$^3$ s$^{-1}$; min: 3.3 – max: 21300 m$^3$ s$^{-1}$) as well as sediment (19; $\mu$
$= 620$ m$^3$ s$^{-1}$; 25 – 2000 m$^3$ s$^{-1}$) but those observations are already associated to large uncertainties. An appraisal of the existing
data and estimates of drained volumes based on satellite imagery from before and after the GLOF events could provide an
approach to estimate the role this large number of GLOFs plays in overall sediment fluxes and could be a focus of future
research.
**4.6 Comparison to previous inventories and limitations**
The most comprehensive databases of GLOFs covering HMA were produced by Veh et al. (2019a), focusing on moraine-
dammed lakes exclusively but identifying many previously unidentified events. Zheng et al. (2021b) also identified many
events that had hitherto gone unidentified. Both studies have been consulted for this study. 144 GLOFs in this inventory (21%
of the total) were previously only reported in (Zheng et al., 2021b), 15 (2%) in (Veh et al., 2019a). We also followed the
approach of Zheng et al. (2021b) to question any previous records and keep a separate record of events that were initially
identified as GLOFs but turned out not to be or cannot be ascertained to be. Future studies should investigate events we have
identified as outbursts from water pockets (22 cases as well as potentially more in what we have discarded as GLOFs in a
separate database), to better understand involved processes. None of these water pockets are visible before any event, and
posterior evidence of their existence is limited to vast subglacial channels that appeared (e.g. Figure 2C) as well as apparent
rapid lowering of glacier surfaces but sometimes simply the observation of previously filled crevasses at the snout where a
debris flow suddenly emerged. In Switzerland, these types of outburst floods have been earlier estimated to account for 30 to
40% of all glacier floods (Haeberli, 1983), but detailed investigations are limited to one especially disastrous case dating back
to the 19$^{th}$ century (Vincent et al., 2015). All events in HMA originated at the terminus of relatively steep glacier tongues and
may have some similarity in genesis to glacier detachments (Kääb et al., 2021, 2018) but runout properties – fast debris flows
with no visible fraction of ice - are distinctly different. More comprehensive investigations into these cases will be required in
future. Veh et al. (2022), a global study that also covers the region already provides an accessible database, however a
considerable smaller number of recorded events (459; 11 events in our database were sourced from this database).
A large motivation for GLOF studies has been the question of whether GLOFs have been increasing in the recent past with a
change in climate. Recent global and regional studies on moraine-dammed lakes have shown that there is no apparent trend,
but suggest a possible lag that may eventually result in an increase in such events (Harrison et al., 2018; Veh et al., 2019a). A

recent global study investigating all types of outbursts suggests a weak coupling of temperature rise and GLOF frequency (Veh et al., 2022). We can confirm findings from Veh et al. (2022) that currently available records likely do not cover all GLOFs that have actually occurred. The fact that we still overlook past events and the strong regional variability seem to make it challenging to find definite climate relations and any such statements should be treated with caution. For some individual lakes that have a documented history of draining since the end of the 19th century (e.g. Hassanabad/Shisper, Kyagar, Merzbacher, Karambar) no apparent trend is visible, while damages have likely increased due to increased exposure of infrastructure in high mountain areas (Li et al., 2022). Hotspots where ice-dammed lake outbursts have repeatedly been unleashed have, in some cases, even completely ceased to exist as the glacier tongues have receded so far that they cannot dam the main valley anymore (e.g. the Shyok glaciers). This is in line with a recent global study finding a decrease in extreme outburst floods from ice dammed lakes (Veh et al., 2023).

The attention that GLOFs in general have received in the region in the past (see Emmer et al., (2022) for a comparison of events versus actual studies globally, with a notable discrepancy for the Himalaya) has possibly led to the misidentification of GLOF events. Flash floods and debris flows where the source remains unclear are often immediately recorded as GLOFs in media (especially in the Western Himalaya, the Karakoram, and Hindu Kush), as this generally gives it more attention than the term 'debris flow' would. Conversely, due to a lack of records in unpopulated areas many GLOFs may have gone unrecorded (Veh et al., 2022; Zheng et al., 2021b) as confirmed in this study. What remains poorly documented so far are impacts in numerous dimensions. Not only are direct impacts poorly recorded in accessible format, secondary medium and long-term effects remain missing from the literature. People injured (qualitatively and quantitatively), people displaced temporarily and permanently, gender and age group of people affected are rarely reported. Effects on mental health have been recently reported but no studies exist in scientific literature (Ebrahim, 2022). With high rates of both erosion and deposition, GLOFs can cause damage to agricultural land as well as infrastructure for years after the actual event, eventually causing damage from the remaining devaluation of land or lack of access to health facilities, markets or education. Information on the severity of deposits would be needed for actual economic impacts and is crucial to further contextualize GLOFs in loss and damage frameworks (Huggel et al., 2019). To this end, it is crucial that future efforts rely on datasets that are transparent and accessible to all stakeholders to make decisions on adaptation that are targeted and sustainable.

**5 Data availability**

The GLOF database is published under https://doi.org/10.5281/zenodo.7271187 (Steiner and Shrestha, 2023). The code to process data as well as the development version, where new events are added regularly, is available on GitHub (https://github.com/fidelsteiner/HMAGLOFDB). Events added in the development version will be quality checked before they are retained in further releases published on Zenodo. To provide an accessible version of the data to non-academic stakeholders, the database is also visualized on an interactive map accessible from the README file in the published database as well as the development version.

## 6 Conclusion

In this study, we present a comprehensive compilation of GLOF events in High Mountain Asia from the mid 19<sup>th</sup> century until 2022. The inventory is machine-readable and version controlled and will be updated as information on new events become available in future. It includes basic information on time and location, involved processes and impacts and is linked to other inventories of glacial lakes and glaciers allowing for future investigations into drivers of outburst floods. Of 697 individual events, 47% have a known month of occurrence, allowing investigations into seasonality, and 39% have a recorded day of drainage, allowing future investigations of prevailing weather preceding the event. 52% of all GLOFs can be associated with a lake mapped in at least one glacial lake inventory, and 95% to a mapped glacier, allowing for straightforward analysis on how upstream glacier mass loss and lake area change influences the occurrence of outburst events. We have a good overview of what type of lakes were the source, revealing that large regional differences exist, with sparingly few ice-dammed lakes in the Himalaya and only very few moraine-dammed lakes in the Karakoram linked to GLOFs. The mechanisms of lake formation or drainage are documented for only very few events. Volumes of floods are not only seldom documented but also highly uncertain. However, the combination with lake inventories allows for approximate estimates, with lake area changes as a proxy. With a record of minimal potential reach of GLOF events in 66% of all cases we can show that GLOF events generally do not exceed reach angles of 15° (and events with higher volumes even less), information that can be useful for future hazard mapping.

Our dataset suggests that 906 deaths were directly associated to a GLOF event, three times as many as previously reported for the region. Compared to other mountain hazards in the region, GLOFs however have caused a relatively small loss of human life. Other impacts however remain relatively poorly documented.

With 7% of all events recorded here never mentioned in literature before coming from local and oral information rather than other academic publications. Our study emphasizes the importance of considering all types of sources and acknowledges the high likelihood, that our current records of GLOFs probably remain an underestimation of actual events. Conversely, we also emphasize the importance of cross-checking sources carefully, as mass flow events in the region have been previously mistakenly recorded as GLOFs without sufficient examination. While this is the first GLOF inventory to comprehensively address downstream impacts, much of this information has never been recorded. Injuries are rarely recorded nor are the monetary values of damaged property or the long-term economic effect of damaged infrastructure. Future studies should evaluate methods to estimate damages rapidly, which will require more interdisciplinary approaches including social scientists for field assessments and economists for ways to upscale risk and damage assessments. Further focus should also be given to documenting local and indigenous knowledge on GLOF hazards, which could reveal impacts that so far have been overlooked. As our understanding of the changing cryosphere in HMA is ever increasing, and with the expansion of our understanding of topics such as snow melt and permafrost, future studies should attempt to combine inventories of lakes and GLOFs with potential processes even further upstream that may result in cascading hazards in future. Such issues are expected to increase

in future. Other hazards should ideally also be documented in formats that make it possible to pair it with already existing
inventories for regional studies and modelling attempts.
**Author contributions**
FS incepted the study and compiled and analysed the database with JFS. FS and JFS wrote the manuscript. RS, YD, SPJ and
SI helped with the initial compilation of the database. AA and SW helped with compilation of data from the Indus basin, KMW
helped with data from Afghanistan and Central Asia and TZ provided input from Chinese events. All authors contributed to
the final version of the manuscript.
**Competing interests**
The authors declare that they have no conflict of interest.
**Acknowledgements**
This study is supported by ICIMOD's Action Area A on managing cryosphere and water risks under Strategic Group 1:
Reducing Climate and Environmental Risks. Action Area A works to address the challenges posed by climate change on
cryosphere, water, and related hazard in the Hindu Kush Himalayan region. ICIMOD and its regional member countries --
Afghanistan, Bangladesh, Bhutan, China, India, Myanmar, Nepal, and Pakistan -- gratefully acknowledge the support of
Norway and Switzerland for funding the activity for this research. The views and interpretations in this publication are those
of the authors and are not necessarily attributable to ICIMOD. The authors gratefully acknowledge the support of Prabhat
Shrestha and Narayan Thapa in the development of an interactive dashboard for the database. We are grateful to Ethan Welty
and two anonymous reviewers, who helped in considerably improving the manuscript.
**Review statement**
This paper was edited by Katrin Lindback and reviewed by Ethan Welty and two anonymous referees.

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
