# Peer review of "HMAGLOFDB v1.0 – a comprehensive and version controlled database of glacier lake outburst floods in High Mountain Asia"

_Earth System Science Data, 2022_

## Referee Comment (RC2)

Dear editor and authors

In this manuscript, Shrestha et al., presented a comprehensive database of Glacier Lake Outburst Floods (GLOFs) in High Mountain Asia (HMA). The authors combined three databases of glacier lakes in this work. The compilation of the GLOF database is largely based on literature review of articles from different sources including peer-reviewed papers, book chapters, technical reports as well as online news articles. The authors also take local knowledge into consideration, which is believed to be necessary considering the issue with under or over reporting of hazards in rural areas in HMA. However, this also brings challenges to the reliability of the sources. The novelty of this database lies in its inclusion of impacts of these GLOFs downstream, which could be difficult to quantify, fact-check and describe in a single .csv file. The authors have also created an interactive map and dashboard for visualization and quick check for non-academic users. I praise the efforts that have been put on compiling such a database, which has great value in creating vulnerability assessments and hazard adaptation plans for mountain communities.

However, I also have a few major comments on how the article is written regarding ESSD guideline and how the data is archived, and quality controlled.

1) First, the authors spent a lot of efforts on analyzing and interpreting the data in Sect. 3 and 4. Of course, an overall statistic of the data could be included. However, I believe, since the article is about presenting a dataset, the emphasis should be put on elaborating the methods used to produce the data, the choices of the variables, analyzing the quality, uncertainties, and limitations in the data and how it could be useful in other studies.

2) The methodology about how the lake dataset and GLOFs dataset were derived are not detailed enough. I understand that the lake data was compiled from three earlier datasets from different years. But it is not clear how they are different in terms of coverage and quality, if the three datasets are merged or used for GLOFs happened in different years separately, and if they are merged what the rule for merging is. For the GLOF dataset, the authors stated briefly that the data was derived by reviewing articles and interviews from different sources and verified by satellite imageries, and that false reported events are removed. But I think more details are need in describing and discussing this process as from this description the readers have no idea how reliable these derived data are. And the removed cases need more vigorous justification. Since this article is about the dataset not the interpretation of the data more discussion could be put in these aspects.

3) In terms of the datasets, the authors indicate that the dataset is publicly available on ICIMOD data portal (https://doi.org/10.26066/RDS.1973283). I assume this will be the main platform for downloading the data. However, the HMAGLOFDB_v1.0.csv file downloaded from there is not accompanied by either a metadata file, a description file or the HMAGLOFDB_removed.csv file. Thus, the user who downloads the data from there has no idea what each column in the data file means. I later realize that those files are included on the Github repository. But in the ICIMOD data port there is no mentioning of the Github repository. It would be nice to centralize these different bits of data or at least link them together to be more user friendly, especially for non-academic users since they are considered as important stakeholders in the article.

4) My biggest concern is the criteria for choosing the mechanism involved in lake breach or drainage. It is not clear to me how the authors decided to adopt a definite mechanism or mark it as unknown. I did not check all the GLOF events in the data files. But for the lakes I checked (next to Kyagar glacier and Khurdopin glacier) there seem to be some issues with this.
The 34 GLOFs of the lake next to Kyagar glacier are well documented in different articles, which are cited in the HMAGLOFDB_v1.0.csv file. The GLOFs before 2018 were believed to be triggered by ice-dam failure linked to subglacial drainage. And the one in 2018 was more likely to be linked to overtopping. However, the entry for the mechanism is unknown. I don't know what the authors' reasoning behind that. The GLOF next to Khurdopin glacier are marked as caused by 'englacial tunnels. However, Bazai et al. (2022) used a subglacial hydrology model to simulate the sudden drainage. Clearly, Bazai et al. (2022) thought the drainage was likely to be linked to subglacial drainage system. The authors of this manuscript have cited Bazai et al. (2022) but decided to adopt englacial tunnel as the mechanism.
5) There are many events that only recorded in 'this study'. We have no idea how they are identified and quality controlled.

Besides, the major issues I also have a few minor comments:

L90 : (Chen, et al., 2021) -> Chen et al. (2021). There are also some other places that the citations are written not according to the convention.

L115: This kind of statement or practice does not seem to be very rigorous. The authors have excluded many cases that might have been caused by debris flows but include cases that have happened far away from any glacier just because they appear in a landscape that was most likely glaciated at one point?

L120-124: The information in this part is not included in the dataset but only describes how the data is analyzed for the discussion in Sect.4. Following my major comment no.1 I don't know if this should be put here or be included at all.

L193-194: In which place did the other 0.8% happen?

L 339-342: Need some references here.

Fig. 2: It is not clear why this figure should be included and why the pictures of these glaciers are selected. These pictures are not a part of the database; or should they be included as a part of the database? Something could be a reference is the Norwegian Water Directorate GLOF map (http://glacier.nve.no/Glacier/viewer/GLOF/en/).

Fig. 3 Maybe it is better to use another colormap to be color-blind friendly.

Fig. 4 A Should the cause of lake appearing be 'glacier melt' or 'glacier retreat'? Or it means something else?

Fig. 6 Not quite sure what the x-axes represent.

---

## Author Comment (AC1)

**CC1**: 'Comment on essd-2022-395', **Jonathan Carrivick**, 12 Jan 2023

https://doi.org/10.5194/essd-2022-395-CC1

1. Considerable amounts of literature, governmental records, international databases and media reports were analysed by Carrivick and Tweed (2016) and they compiled number, volume, discharge, and damage and societal impact quantities (e.g., they report at least 6300 deaths from GLOFs across Himalaya) where possible and for hundreds of events across the Himalaya. They concluded the societal impact of GLOFs is greatest in central Asia of any world region. Their datasets are freely available. So, I really think you need to compare to, and refer to their study and its findings!

Carrivick, J.L. and Tweed, F.S., 2016. A global assessment of the societal impacts of glacier outburst floods. *Global and Planetary Change*, *144*, pp.1-16. https://doi.org/10.1016/j.gloplacha.2016.07.001

**Thank you for your comment. The study by Carrivick and Tweed (2016) was instrumental in the development of the HMA GLOF database and has contributed over 5.4% of the database recorded events.**

---

## Author Response (AR1)

Dear Editor,

Thank you for your patience. We have now completely revised both manuscript and database to follow suggestions of Ethan Welty (specifically on IDs and Metadata) and the two anonymous reviewers. To keep track of the changes we naturally provide the marked-up document but also list the most important changes with respect to the individual comments below.

CC2 (Ethan Welty):

When responding to Ethan Welty's comment we originally were skeptical of the suggestion to have only a running ID from 1 to infinity but have now gone with the same, avoiding further confusion with the GLIMS identification. We now also provide a MetaData file that is fully machine but also human readable via the .yml format.

Reviewer1:

We understand that the different DOIs have caused quite some confusion, and hope that the more streamlined way now executed makes sense (L684 and L687). As noted also in the initial response after submission, we believe that making it as accessible as possible for both academics as well as non-academic stakeholders is crucial.

We understand the concern regarding finding evidence on satellite imagery and have followed the reviewer's suggestion by adding the column *'Sat_evidence'* that now provides details on the imagery used to confirm an event that this study identified. We have done this for three events not previously recorded in any literature, reports, or social media, where additional evidence was crucial. Future content updates to the database could see this done also for other events that need confirmation of the occurrence. We have taken this chance to revisit many of the events. For the other three events where it was simply not possible to find the respective evidence, we have now decided to remove them from the database. Conversely, the reviewer noted that we were very conservative on some events from Merzbacher lake, which we have now carefully revisited, and have added 19 of these events to the database. This leads to a new total number and eventually a slight change of relative statistics throughout the manuscript (but no change of any of the conclusions).

We understand that the reviewer was concerned regarding our reasoning on why we collected certain variables. We have hence now considerably expanded on this in Section 2.2, without making the manuscript longer as the complete discussion of lake inventories (including one figure) was removed, following comments by all reviewers.

We were originally not entirely convinced by the reviewer's suggestion to add full citations into the database itself and in the review response hence also just refer to our original document with full references. We have however since come around to also adding these full citations separately, documented in the Metadata and readable without causing the complete clutter we were concerned about. This eventually has however now also led us to remove the original word document, to make the final database clearer (only leaving two .csv files as well as the Metadata file).

The reviewer had valid concerns about how we deal with the inherent uncertainties of the data we collected. We now address this in more detail in sections 3.1 and 4.6. As we have mentioned in the text, treating uncertainties collected from interactions with stakeholders during fieldwork looks different than for cases collected from satellite imagery. We have however also followed an approach,

where if we cannot be sure that the event was a GLOF, we remove it from the database or do not record it as such in the first place.

Regarding the concerns related to the analysis we already show in this manuscript, we hope to have dispelled the concerns. We have removed large parts of the discussion with regard to lake databases and agree that this was not helpful for the manuscript. We have also provided better reasoning (L538), including added citations especially on studies that so far do not have access to actual runouts of GLOFs, that hopefully make clear why collecting this data (and reporting its uncertainties) can be helpful for future investigations.

Revierwer2:

The main concerns of reviewer 2 match with reviewer 1 and are addressed above, namely that we put too much focus on analysis and too little on data retrieval explanation as well as the confusion on the location of the data. Additionally, there was concern about us having misidentified drainage mechanisms for a few cases (Kyagar, Khurdopin) that caused many GLOFs. This was indeed the case, and we regret the error. We have updated this in the database and as a result there was a considerable change to Figure 4 as a result.

Kind regards,

Finu Shrestha

On behalf of all co-authors

**CC4**: ['Comment on essd-2022-395'](), Ethan Welty, 02 Feb 2023

Dear Ethan,

thank you for a careful reading of the manuscript and the immensely helpful suggestions made, greatly appreciated. We have addressed each of them individually below, with your comments in Italics followed by our responses in bold.

*1. I would advise against adopting an identifier of the form GF{longitude}E{latitude}N_{counter}. It may at first seem like a convenience to build in spatial coordinates into an ID, but what if the coordinates are later changed? Then either the ID has to be updated (please never do this) or the coordinates in the ID no longer match those in the table (which can also lead to confusion). Furthermore, at least in the case of GLIMS, it has led to people generating their own "GLIMS" IDs which don't actually exist in GLIMS (e.g., "one day, when I submit my data to GLIMS, it will have this ID").*

**We agree that GLIMS IDs can change depending on who determines the location on the lake. However, compared to using the lake name (majority of lakes lack names) or the conventional method of numbering lakes based on their position relative to the major stream in a clockwise direction, this method offers a simple and convenient way of identifying and locating specific lakes. If we go with a name identifier we are faced with having to call many 'Unknown' or even worse are faced with the added confusion of numerous transliterations of local names. A running ID does not seem feasible considering the database will incrementally increase. We are also not sure why coordinates should change in the future. Each GLOF event is a separate historic event. If the location of the lake shifts considerably, it will be a new lake and hence a new ID.**

**We agree however that suggesting that this is a GLIMS identifier is not helpful, as it confuses it with the original database and we do want to suggest that what we provide here should be in future associated with GLIMS. We have therefore removed the GF at the beginning, and call it a GLOF ID (you could also see it as a random set of alphanumerals, with the added benefit of holding some location information). We have also adapted the phrasing in the manuscript.**

**Addressed: In L259 table and database file. (When responding to Ethan Welty's comment we originally were skeptical of the suggestion to have only a running ID from 1 to infinity but have now gone with the same, avoiding further confusion with the GLIMS identification).**

*2. Looking at Table 1, it seems like, since each lake can have multiple GLOFs, that the database would benefit from being split into two tables: one for lakes and one for GLOFs? This is less a concern if the database exists in a split (i.e. "normalized") form, and the tables are joined into one for publication, since the underlying structure ensures that lake attributes are always the same for all GLOFs associated with that lake. But if the data is maintained as a single table, this consistency is not guaranteed.*

**Reply: The database aims to provide additional information beyond the frequency of GLOF occurrences. Multiple GLOFs from the same lake can have varying impacts based on the degree of their drainage, so only the socioeconomic factors and outburst date will undergo modifications in the attribute, as you rightly pointed out. However, and this may have been confusing on our part, we should clarify that the paper really only attempts to characterize GLOFs, not lakes. The IDs of lakes, used in also in databases that are not produced by us are placed in the database for GLOFs to make a direct association with GLOF events possible, where we can ascertain the source. Also following suggestions by reviewers, we now removed the part describing the lake databases as well as the associated discussion, focusing specifically only on GLOFs and hope that this clarifies the intention here.**

**Addressed: L264 (removed the lakes database part)**

3. You write that HMAGLOFDB_Metadata.txt is "machine-readable". Certainly, a machine can read each character of a text file, but what matters more is that the content of the file follows a standard format. Is it JSON, YAML, XML, …? The .txt file extension suggests that it follows no such convention.

**The metadata file so far was indeed not in .yml or .json. Following this suggestion and the revision of the database, we have however decided to write it into YAML format and have updated this in the new database.**

**Addressed: L169**

**Reviewer comment on essd-2022-395**

**Anonymous Referee #1**
Referee comment on "HMAGLOFDB v1.0 – a comprehensive and version-controlled database of glacier lake outburst floods in high mountain Asia" by Finu Shrestha et al., Earth Syst. Sci. Data Discuss., https://doi.org/10.5194/essd-2022-395-RC1, 2023

Dear editors,
Thank you for giving me the opportunity to review the manuscript essd-2022-395 "HMAGLOFDB v1.0 - a comprehensive and version-controlled database of glacier lake outburst floods in high mountain Asia" by Shrestha and co-authors. In this study, Shrestha et al. compiled an inventory of glacier lake outburst floods (GLOFs) from sources such as scientific literature, media, and eyewitness accounts since the mid-19th century.
Of the 682 documented GLOFs, 49 cases have not been previously recorded in other databases and appear for the first time in this compilation. The authors define dozens of variables that characterize the location, size, impact, and consequences of these GLOFs and attempt to gather all available information to populate these attributes with numerical or descriptive information. An important observation is that reporting on GLOFs has not been systematic in recent decades, leaving large gaps in the database that could be filled by further research.

Shrestha et al. report on the largest GLOF database to date in High Mountain Asia. The authors validate each case using geomorphological evidence downstream of the lake (but not necessarily changes in lake size) on satellite imagery, as well as assessment by local experts. In my opinion, this assessment is among the most carefully prepared inventories in this region, and I commend the authors for this work. The authors also provide a list of previously reported GLOFs that they excluded from this database, and the reasons for excluding these cases (e.g., incorrect coordinates or little evidence of former lakes) sound largely reasonable. I also appreciate the interactive map that allows non-experts to have easy access to this data (but without a download option).

Despite these efforts to compile this database, its presentation and associated manuscript require thorough revision, not least to comply with Earth System Science Data guidelines.

I will first comment on major issues associated with the database, followed by issues in the manuscript.

**Dear Reviewer,**

> **We are extremely grateful for the careful reading of the manuscript and the detailed comments provided. Specifically, regarding the visual dashboard, there is indeed no direct download option, as eventually the dashboard should point to this publication. This is a bit of a chicken and egg problem and with the already two different databases we point to we have decided to remove the link to the interface for visualization. We will keep developing the dashboard further and any data that will be updated for the database will also be updated on the dashboard, but the link will only be visible in the database itself (on github as well as RDS).**
> **We will address each of the concerns raised individually below with your review in italics followed by our response in bold.**

*1.) The database is currently archived on at least four different platforms (Zenodo, Github, ArcGIS and ICIMOD), and some files are available on one platform while missing on others (e.g. HMAGLOFDB_removed.csv or the list of references is not on Zenodo). I strongly encourage the authors to use ONE non-proprietary, accessible platform, i.e. without the need for a login account, to deposit their data in order to pursue their goal of presenting a version controlled database. This repository should contain all files contributing to this database (e.g. metadata, the list of references). My preferred choice would be Zenodo, but I leave that decision to the authors.*

**We agree with the reviewer (as with others who made similar comments) that this is not ideal. Regarding github/zenodo, these are not two repositories. Github is where the data lies and is updated, zenodo is linked to github and where any new stable release is published. RDS is the regional database system that allows us to reach stakeholders well outside of academia, and we hence believe that this is a crucial location as well. It however has the disadvantage of requiring a log in etc. We hence will now provide just one DOI for the database (for the zenodo location), which will include the link to the dashboard as well as the RDS database and will refer to the github page as the 'development version' as has been done previously** (e.g. Welty et al. 2020)**. The description of the text is adjusted accordingly, and it has been added to the abstract.**

**Addressed: L29-L30.**

*2.) The references column should contain the full name of the reference (author, title, journal, year) to make the underlying source easy to find.*

**We would like to refer to a full reference document provided separately in a text file (HMAGLOFDG_CIT) where the full citations are given. We now provide one column with full citations of all papers sourced ('Ref_scientific_full'). Adding separate columns for authors and titles, especially when there are multiple sources makes the table extremely clumsy (and would mean we would need to keep adding more and more columns as time goes on) and we would hence prefer to keep this information separate.**

**Addressed: L259 (table) and database file.**

3.) This database contains dozens of new cases that have not been reported before. I believe it is important to properly document these new cases, e.g., with satellite imagery before and after the GLOF showing lake area change or downstream impacts. I would like to encourage the authors to provide supplementary material for these cases.

**Providing satellite evidence for all is not possible for different reasons. (a) Some events are based on local information from stakeholders we work with predating the satellite age. And while we did check for all cases whether a GLOF would be possible (i.e. there needs to be a lake or a depression that could have been a lake in the past) and locals did not confuse it with debris flows from heavy precipitation or snow melt not for all imagery is available (i.e. before and after documentation). (b) In some cases (especially in the Upper Indus basin) there is no evidence of the lake as we suspect them to be englacial lakes (or water pockets as we now call them, we have adapted that throughout, also considering that in some cases these pockets may be small or scattered and hence lake becomes a misnomer). Again, we checked all these cases and have discarded events where it is highly likely that it was just a debris, also based on the local information of the process chain. Following the suggestion of another reviewer, we have now also taken greater care in**

describing this process of data collection. We realize that a lot of information on the events we here present the first time were missing from the actual database, which we have now added in the separate column on Ref_other. This mostly refers to information gathered locally while being present for first responder missions by co-authors of the manuscript.

For cases where it was available and where we did consult satellite imagery because local evidence was not available, we now provide an indication of the imagery where the evidence was collected in a separate column. Note that many of these events we could not associate with any particular date and have happened likely before satellite coverage started.

Addressed: L259 (table) and database file.

*4.) The value of this database is that it can be used to calculate trends in GLOF occurrence, hazard, and risk. To this end, each case should be provided with at least an approximate time stamp. If the exact date is not known, the authors could at least indicate that the GLOF occurred before the first available satellite image (Landsat or Corona) was acquired or during a period embraced by two satellite images. This information should be added for the entries in lines 474 to 660 in the current database.*
*For these cases, it is still difficult to assess whether the occurrence of these cases could be more accurately dated, e.g., by using satellite imagery that cover the outbursts or not. In addition, it would be good to split the Year_approx column into two columns, one with the latest possible date (or year) before the GLOF and one with the first possible date (or year) after the GLOF.*

Also following our response to the previous question, we would like to emphasize that the database we provide is not remote sensing based but relies primarily on field evidence, additionally to cases previously documented in literature. We believe this to be crucial as it allows us to make statements on impacts, but we also believe that this local knowledge so far has been relatively less recorded in literature. Since our co-authors are specialising on this kind of fieldwork, we are also able to discern potentially false information from actual GLOFs. We do however use satellite imagery for cases where no other way of supporting our claims is possible. We now have introduced a column called Year_Sat for those events that have likely occurred in a period covered by satellite imagery, noting the earliest year we could find that provided evidence for it. For many cases Corona or L7 are actually not adequate as the events were too small to be identified. In these cases, we rely on available Maxar imagery (Google Earth) or Sentinel imagery (see the remarks column). This still leaves us with many events that have no date, which are generally events that have been documented locally but there is no satellite imagery. We still believe this information to be valuable as a density of GLOF events, even without a time stamp (but say an occurrence during a century) tells us something about their role in forming landscapes.

Addressed: L259 (table) and database file.

*5.) I have checked some GLOFs (yet not all systematically) and some cases seem to have limited evidence on satellite imagery, or the coordinates seem to be in wrong locations, e.g. at Langco, Tulaco, Phyang, Langbu Tsho or Bugyai. Oubuguoco has two entries, and I wonder if this is a duplicate? The GLOF from Pogeco in 2002 appears to be a misidentified GLOF, according to Nie et al. (2018). Looking at the removed cases, I wonder why so many GLOFs from Lake Merzbacher were discarded? Ng and Liu (2009) and Kingslake and*

*Ng (2013) provide a thorough compilation of these cases, including flood volumes and peak discharges.*

**Thanks for checking through the database, we respond to the individual issues one by one. There is only one entry for Oubuguoco that we can find. We have rechecked the Lang co event and it is confirmed and matches with the reference cited (coordinate shows the lake). The Tulaco coordinates are also confirmed with the reference cited, the coordinate is located on glacier as lake type was supraglacial. The Phyang case has been poorly documented by local sources (see the added link in the sources given) but gives some confidence on there having been a possibly small lake. The coordinates where of the impact area however and have now been adapted to the source area. For Langbu Tsho the coordinate is correct and confirmed with the reference cited. For Bugyai the coordinates were shifted, which has now been corrected and relocated to the lake. As for Poge Co, there seems to have been previous confusions over names (see (Nie et al. 2018) for a detailed discussion on the case) – the event from 2002 is the same as Dega Co. Note that however that lake's name is also used for two different cases, and they can be differentiated by their ID.**

**As for the Inylchek/Merzbacher cases we were concerned that many of the recorded events were very close together and, with experience from many other GLOF cases it is hard to know whether there is not only a mix up in dates or it was the same GLOF, only with release of water over multiple dates as the channel expanded and hence events were recorded multiple times. We therefore followed the Glazirin compilation (Glazirin 2010), which based on the local insight we considered most appropriate until the date of its publication. Upon revisiting the removed cases we agree however that some of the years covered in (Kingslake and Ng 2013) do not appear at all and we have revised this accordingly and brought those into the database (e.g. the 1968 case). This has of course then also resulted in slight changes to the overall statistics (as have the changes noted above for few cases on coordinates and hence elevations) however due to the small number they do not change the overall quality of the database.**

**Addressed: L373 (table under outburst occurrence column for Merzbacher GLOF).**

Regarding the manuscript, I have the following concerns and recommendations for improvement.

*6.) The reason why the variables were selected could deserve a much stronger motivation/ justification. Why is it important to document river basin and lake volume?*
*Why is it important to distinguish between female and male fatalities from GLOFs? In this regard, the metadata table should also be part of the main manuscript and adequately explain the meaning and units of each variable in separate columns.*

**We have now added more information on our reasoning behind the variable choice in Section 2 in detail. The coordinates of the GLOF's source as well as the final impact location downstream are provided as these coordinate supports to prepare a hazard zonation map to analyse and evaluate the associated risk (Uddin and Matin 2021). The source lake location (IDs), associated glacier, lake volume before flood, flood discharge and river basin are given where known and existent. This information is crucial for planning, designing and implementation of large-scale projects like hydroelectric power plants and other types of infrastructure, in order to ensure sustainable**

development. Information on impacts on livelihoods and infrastructure are provided to conceptualise flood-induced coping mechanisms, enhance livelihood security, and foster self-reliance toward economic stability (Ministry of Planning Development & Special Initiatives 2022). This information is presented in quantitative as well as qualitative formats enabling the database to be read by machines, while retaining information that may not be readily quantifiable. Information on fatalities is categorized by gender and disabilities, as disasters impact women and those with disabilities differently than men (Erman et al. 2021). This data is important for addressing gender inequality, cultural beliefs, and socio-economic factors, as well as advocating for the integration of gender and disabilities perspectives into disaster risk management efforts (UNESCAP 2022).

**The name and explanation of each variable and units are already provided in Table 1.**

**Addressed: L104.**

*7.) I feel that the compilation of lake data distracts from the real topic of this manuscript, which is the compilation of GLOFs. One motivation for the authors seems to be that they want to add a unique ID to each GLOF according to a previously compiled lake inventory (is my assumption correct?). However, the number of lakes may change over time, and the inventories have different minimum mapping units, different criteria for mapping lakes, or choose different distances from glaciers to consider a lake a glacial origin or influence. The methods used to map and categorize glacial lakes are subject to very different challenges and uncertainties than the creation of GLOFs and are therefore part of a very different story. It is therefore surprising that the results begin with a presentation of lake area change, given that the title and scope of this manuscript refers to GLOFs. There is also little information on how these lake inventories were mapped, particularly the multi-period inventories (2000, 2010, and 2020). Were these inventories prepared by the authors or as part of a previous study? That being said, I strongly recommend removing the lake inventories from this study and focusing on the GLOFs.*

**Our motivation was indeed to have this closely linked as lakes are of course the source and it would allow a quick combination of the datasets for further analysis. Following your and other's comments we appreciate however that presenting two datasets in one manuscript dilutes the focus completely and have hence decided to remove this part. Of course, the associated lakes will remain in the database, so will some of the discussion on what the value of linking them is, but the actual discussion of the databases itself has been dropped completely.**

**Addressed: L264 (removed).**

8.) Removing the lake inventory from the study would also help avoid comparing lakes with and without outbursts (e.g., in Figures 5 and 6). ESSD is a data-driven journal and discourages scientific interpretation of the data. Instead, the discussion might focus more on the completeness (or gaps) of the variables. Could these be closed in future updates of the database? If so, how? Is there a need to collect additional variables? If so, which ones?

**We absolutely agree with this assessment and have dropped these figures. We retain some discussion on the resolution of the lake inventory which goes in the direction, why recording GLOFs from local sources matters as remote sensing alone cannot capture every type.**

**Addressed: L119-L122; L130-L135.**

Specific comments (line by line):

L1: It is difficult to say why this is version controlled. Have there been previous versions of that database?

**This is the first GLOF database prepared for HMA. The database will be updated annually in collaboration with regional partners, and the updated version will be released at the end of each year. The advantage of the github repository (as for previous ESSD publications like (Mankoff et al. 2021)) is that any updates can be traced in future. This would also make future efforts of tracing how some events made it into records simpler. From the review we have experienced how confusing some records are because they are repeated from secondary literature and at one point it is not clear anymore where the records have been initially documented. We also think this is crucial for the region as it helps to build trust in data and hence the sharing of the same.**

**Addressed: L684-L690.**

L22: A number of cases seem to have occurred before 1833 according to the csv file?

**Yes, those events are very limited (and no trends etc should be read into it) but it has been mentioned in line no. 189 of the original submission.**

**Addressed: L301.**

L27: It is common practice in ESSD that the abstract ends with the link and a reference to the repository. Please add.

**We regret this omission; this is now stated at the end of the abstract.**

**Addressed: L29-L30.**

L29: 'expanse': amount?

**Yes – and for a spatial domain we believe that should be appropriate wording.**

L31: 'on a large scale': please be more specific.

**Added detail for the region.**

**Addressed: L34.**

L32: 'Many of these lakes': better use another phrase. According to your database, about 1% of all lakes had outbursts; 'results': please check the verb here.

**Revised with 'numerous lakes' and 'resulted in'.**

**Addressed: L36.**

L34: 'many decades in different parts' please be more specific.

**Revised with 'GLOFs have been recorded in various parts of the world for decades' – the details follow from the citations following.**

**Addressed: L37-L38.**

*L35: 'HKH': please explain.*

**Revised.**

**Addressed: L40.**

*L39: 'n=…', just say 2916 lakes? Did these two studies use exactly the same methods, i.e. same mapping area, same buffer around glaciers?*

**Thanks, revised. Yes, they both apply buffer area within 10 km of glacier extent for glacial lake inventory. However, since we now remove the discussion on lake inventories, this discussion has also been removed.**

**Addressed: L43-L44.**

*L45: 'potentially dangerous lakes': I feel that this phrase adds little objectivity to the discussion of potential changes in GLOF hazard or risk and suggest to remove it.*

**Removed and replaced with hazardous lakes, this follows the papers cited which focus mainly on lakes that hold hazard potential.**

**Addressed: L49.**

*L50: 'strain dams': if they are overtopped?*

**Not exactly. The pressure created by the rising water level can weaken the moraine's stability and cause the surface to erode, which could lead to the dam's failure. We see this more frequently than the actual overtopping due to an excessive water level.**

*L51: 'seismic events': you mean earthquakes?*

**Yes, predominantly but can also be other seismic shocks that may be caused e.g. my nearby mass movements. Hence 'seismic' remains inclusive rather than just focusing on earthquakes.**

*L56: 'in the shadow': you mean downstream?*

**Yes, and changed.**

**Addressed: L60.**

*L77-92: As described above, I discourage from putting the compilation of lake inventory into this manuscript. If the authors would like to keep it, much more information is needed on how the lakes were mapped.*

**Following the response above and the advice given, we have indeed decided to remove this section and limit it only to the acknowledgment that these inventories exist, are tremendously useful and what their drawbacks may be in light of use with GLOFs.**

**Addressed: L88 (Removed).**

*L103: 'overreporting': maybe use another phrase such as 'misidentified' / 'confused'?*

**Revised and replaced with 'misidentification'.**

**Addressed: L123.**

*L107: What about changes in lake areas and exposed lake beds? Is this not a criterion for identifying GLOFs?*

**Yes, they are significant, and we took them into account in this study, when double checking potential sites from satellite imagery. The text has been revised.**

**Addressed: L127.**

*L112: This file is only on Github, as far as I can see. Please choose one repository that contains all data.*

**We regret the omission. As we now only refer to one DOI, all data is available at the location. Additionally, the extra files are also available on the RDS.**

**Addressed: L687-L689.**

*L126-135: The content in these sentences is largely a repetition from the introduction. Please merge this information with the introduction and delete here.*

**Thanks for noting this redundancy. We have shifted some of the material and integrated it in the Introduction. Section 2.3 was furthermore completely revised.**

**Addressed: L151.**

*L134: 'Multiple datasets': please cite.*

**Cited.**

**Addressed: L245.**

*L135: dh/dt: please explain.*

**Rephrased and citation added.**

**Addressed: L44-L45.**

*L136: 'in .git on a rolling basis': please explain.*

**We have now expanded the explanation here.**

**Addressed: L250-L256.**

*L137: 'RDS / DOI': please explain*

**Revised the text.**

**Addressed: L256 and L689.**

*L138-140: The concept of this ID is very confusion: What does GF, E, and N stand for? How do represent the precision of a given coordinate, i.e. how would you write a coordinate of 79.200239? Doesn't Z need to have two digits, if a lake bursts out more*

*than 9 times, as reported in Table 3?*

**We are sorry for the confusion and this format has also changed following another reviewer comment. The acronyms have been updated in the text. The precision is measured by the three decimal places in the latitude and longitude (e.g., GF075474E36344N_3) and has been updated in the document. Z represents infinity but begins at 0. It has now been clarified.**

**Addressed: This part has been removed and updated as per the suggestion by CC2. Please see L171-L175.**

*L144: Please add the entire reference to the database, not only author and year.*

**We would like to point out that all details on references are provided in a separate document (HMAGLOFDG_CIT). We now provide one column with full citations of all papers sourced ('Ref_scientific_full').**

**Addressed: L259 (table) and database file.**

*L146-147: add the name of this file here.*

**Added.**

**Addressed: L236.**

*Table 1: There is hardly any information on why these variables are part of the database. Please add more reason why you selected these variables. Please copy the information from the meta data file here. How did you extract Elevation [m a.s.l.]? How did you obtain Area? Did you map the lakes? 'Displaced_disabilities': missing 'l'; column 'Certainty' is not part of the database, as far as I can see.*

**We have now revised the section around this Table and the description of why and how these variables were determined. 'Certainty' refers only to removed cases and does only show up in that respective table.**

**Addressed: L259 (table) and L151 (section 2.2).**

*L155-166: As discussed above, the presentation of a lake data base is questionable, and I would encourage the authors to remove it.*

**As described above, this has been amended.**

*L172: 'ICIMOD': copy the link here and avoid foot notes.*

**Revised.**

**Addressed: L282.**

*L177: I couldn't see the persons. Please add a circle.*

**We added a marker to visualize the size of the people in the photo.**

**Addressed: L285 (figure 2)**

*L178: 'Milad Dildar': is this the photographer? Do you have permission to use the photos?*

**Milad is one of our field staff and the picture was taken during a field visit of our organisation. He provided the field photograph. The photo credit has now been mentioned.**

**Addressed: L285 (figure 2)**

*L183: 'water line': I couldn't see it. Please add an arrow*

**Added.**

**Addressed: L294 (figure 2)**

*Figure 3: Please avoid red and green in figures at the same time, suggest using a viridis color scale: https://cran.r-project.org/web/packages/viridis/vignettes/intro-to-viridis.html; which study is the source of these 7 regions? Please add more space between the decades/ months. It is challenging to say when decade / month ends and the next begins. Would also be good to report the total sum per decade.*

**Thank you for these comments, completely revised.**

**Addressed: L299 (figure 3); L146-L150**

*L190: I would welcome very much if the authors add a supplementary file that validates the occurrence of these newly reported cases.*

**As discussed further up, we are not able to do the satellite image validation for all the cases, be it for time of imagery availability vs when the event happened, available resolution and for all of the events in the database the simple lack of time. However, we now do supply that for cases where we only had imagery as a source to begin with. We specify the imagery dates where it was visible in the 'remarks' column.**

**Addressed: L259 (table) and database file.**

*L193: Is this statement referring to the removed cases?*

**No, this number was referring to the actual database, we have now amended the text to make this clear.**

**Addressed: L306.**

*L200: allowing 'for'?*

**Revised.**

**Addressed: L314.**

*L207: 'supersaturation': unclear, please explain.*

**Maybe supersaturation in this context was not ideal – this is water accumulating in crevasses (similar to what you would have in surges or what is hypothesized to stand at the beginning of glacier detachments) that leads to eventual rapid drainage. It's neither a lake nor a detachment (the ice doesn't fail all the way to the bedrock) but a superficial event. We now called it 'accumulation of meltwater'.**

**Addressed: L322 (Removed).**

*L209: Why not simply calling it a water pocket, similar to the study of Haeberli, 1983? 'Glacier outburst' sounds like if parts of the glacier get mobilized.*

**We agree that the terminology may be misleading. We have now referred to this here as well as in the database and the rest of the manuscript as 'outbursts of water pockets'.**

**Addressed: L320.**

*221: 22 'GLOFs'.*

**Revised.**

**Addressed: L337.**

*L224: 'inventories': which ones?*

**RGI 6.0 preceding this text, we now adapted it to just say 'this inventory' as we can not proof it for others, but generally believe this is likely true for other glacier outline datasets as well.**

**Addressed: L340.**

*L226: 'this data': change to 'these' (data is plural)*

**Revised.**

**Addressed: L342.**

*L228: How likely is it that all the fatalities came from the GLOF itself? The GLOF was part of a large rainfall event with many debris flows (see also the image in Allen et al., 2016).*

**Rainfall-triggered disasters are often characterized by a variety of hazards that can lead to multiple fatalities, which may not be attributed to a single event. In the case of the 2013 India GLOF event, we have mentioned that 6000 fatalities were caused by a multitude of factors in a complex compound event. Also, mentioned in L398-399. There is no attribution study (and that is likely impossible for the event) but from the way deaths were recorded in Kedarnath (including people who died well outside the flood path but simply**

in the region due to the rains and ensuing floods/debris flows) as well as the intensity of the major driver (rainfall) we believe that actually only a fraction of this number can be attributed to the GLOF. And as we show here, this total number accounts for 86% of all fatalities in the region. Since such numbers matter in the public discourse on a hazard we think it is important to be cautious here since the story looks very different when we say '7000 people died in GLOFs' vs '1000 people died in GLOFs' and that eventually drives investment in hazard mitigation in one direction that maybe should not be the only focus. This is naturally true for any other events as well. This reflection on policy implications of hazard/risk data however goes beyond the scope of this data paper.

**Addressed: L344-L345.**

*L254: 'potentially': why? Did these floods cross the borders or not?*

**Only a few cross-border events have been reported (see next sentence). We have now revised this part to make this argument clear.**

**Addressed: L370-L372.**

*L275: 'hundreds': suggest to add 'to thousands of meters'?*

**Revised.**

**Addressed: L394.**

*L277: 'SRTM': please explain, and mention this part of the work also in the methods; 'between 10 and 50m': how do you know about these uncertainties?*

**This was based on the varying elevation values when selecting elevation from different parts of a lake via the SRTM but is indeed quite handwaving and we did not systematically assess this. We have hence rephrased this whole part together with the following sentence in the manuscript.**

**Addressed: L396-L399.**

*L278: 'errors': of what?*

**See response above.**

**Addressed: L396-L399.**

*L280: 'satellite imagery close before the drainage date': it's still unclear whether (or not) you mapped the lake area before the GLOF in this study. Please be more specific. 'further analysis': which analysis?*

**We did not systematically assess areas of lakes from satellite imagery. For many cases imagery is not available just before the event (to be sure that it didn't grow further between image acquisition and drainage) and we rather relied on the lake inventories that have been already developed systematically (and hence can be compared as well, not further increasing the uncertainty because of operator bias). The areas we report are values reported in the respective sources or if the event has been recorded on our behalf, we did try to confirm the lake area if available from within a few days before the event. We have rephrased this section now to make this clearer.**

We have removed the sentence with 'further analysis' as we have removed this part from the manuscript altogether.

**Addressed: L411.**

*L289: 'wilful tampering with data for political reasons': interesting thought, please elaborate.*

We have now expanded on this topic, based on our experience from fieldwork. This is also true for discharge values for example, where in the subcontinent the misrepresentation of volumes (stemming from taking the originally reported cubic feet per second as cubic meters per second) is frequent and generally inflates the imagined or expected impacts.

**Addressed: L423-L430.**

*L289: Please add more information on how you deem a source of information 'generally trustworthy'.*

The events we report here that have previously not been part of literature are predominately based on fieldwork by co-authors, i.e. interaction with people impacted and visit of the respective event sites. In some cases, the sources were visited, sometimes repeatedly (e.g. the events in Afghanistan as well as most of the events in Pakistan) but during the discussion with locals as well as from local news reports often more events appeared. For those we checked whether the characteristics of a GLOF typical for the site (e.g. for subglacial drainage an initial decrease in flow followed by a steady increase, while for moraine dammed cases a rapid peak) were met and if people were aware of the lake in the upstream changing its properties (often known from herding at high elevation). The most challenging part is to make sure that it was not a debris flow without a lake in play. If satellite imagery can not confirm it. This is where satellite imagery comes into play, but it leaves cases where lakes were too small to be visible on e.g. Landsat (see our discussion on that) or simply not available yet. If people would not confirm that these were indeed lake outbursts, we disregarded the case (also because debris flows are generally more common). We have now added to the discussion in the manuscript regarding this issue.

**Addressed: L639-L685.**

*L305: 'susceptibility': unclear how you estimated susceptibility. Please revise.*

This section has been removed and only parts retained.

**Addressed: L450 (removed).**

*L305-306: Is there any hypothesis that says that lakes at higher elevations should be more prone to outburst?*

As above, this section has been considerably changed. There is no such hypothesis and maybe this phrasing suggests it. In the retained text we have removed this as it doesn't add value to the statement following in the next sentence (that GLOFs are more frequent at lower elevations).

**Addressed: L450-L453 (removed).**

*L306: 'less likely': I guess this is not a probabilistic assessment, so please avoid; 'a larger number of GLOFs happened [AT] low elevations'.*

**Removed completely, see point made above.**

**Addressed: L451 (removed).**

*L320-321: interesting thought regarding the monitoring of debris flows. Is there any reference for that?*

**Unfortunately, we do not have hard evidence for this hypothesis, but this rather stems from the visible amount of literature (in Russian often) from the time before Independence and an eventual decrease afterwards. However (Medeu et al. 2019) argue that at least for Kazakhstan this can be attributed to successful mitigation efforts and we have added this line of thought here now as well (which remains impossible to prove).**

**Addressed: L471-L472**

*L326: could be backed up with more references.*

**We have now elaborated with a focus on the KKH and the reference that discusses this development.**

**Addressed: L477-L478.**

*L327: 'increase of [reported] events'*

**Revised.**

**Addressed: L479.**

*L327-328: not exactly sure how the trends in the reported GLOFs and the research activity fit together. Please elaborate.*

**The increased interest has raised the bar to find and document more events, which we believe is linked – as there was more interest/funding available especially regional researchers wrote more detailed studies on individually events which eventually landed on the radar of the global community. Many of the events we document here for the first time for example were well known in the communities but had simply not been picked up by scientists to model for example and hence resulted in them not appearing in any previous records compiling events. We have rephrased the sentence.**

**Addressed: L480-L482.**

*Paragraph 4.3: Suggest to avoid this discussion. ESSD is a data-driven journal, and too much science can be a reason for rejection.*

**We agree with the reviewer and have completely removed this section, including the Figure.**

**Addressed: L510-L529 (removed) and L545 (figure removed).**

*Figure 6, second panel: shouldn't lakes with outburst have negative lake area change?*

**This figure has been removed. Note however that these are the overall changes over decades. Hence this also includes area changes that may go beyond a GLOF, i.e. the lake could be 10 m2 at the beginning of the period, drain in between completely but refill again to 20m2 due to increased glacier melt or other contributing factors.**

**Addressed: L545 (figure removed).**

*L365: 'risk': you mean hazard (i.e. probability of failure)?*

**Yes, revised.**

**Addressed: L555.**

*Figure 7: Honestly speaking, I cannot see the added value of this figure. 'PZI/ RGI': please explain.*

**We believe that the figure is helpful in visualizing how this database can be combined with other datasets. We have however expanded on this now in the Discussion to make it clear. PZI is also explained.**

**Addressed: L553-L559; L567-L568.**

*Figure 8: Please elaborate how you obtained the Fahrböschung. What is the overall reason behind this analysis? I could not find any motivation in the introduction for this analysis. Why do you plot the data from Kääb et al. (2021) here? Please add axis labels to the inset and describe in the figure caption.*

**As for Figure 7 we believe this Figure is useful to show the potential of having both source as well as impact data available for the whole region for many events, also since this was not possible for previous inventories and is crucial for risk estimation. We agree however that the discussion of this was lacking and have now elaborated on the concept and why it plays a role here. It follows the same argument as in other studies (like (Kääb et al. 2021)) to evaluate a pattern in reach of these events.**

**Addressed: L578-L598.**

*L388: 'mean reach angle': all previous statements refer to the median?*

**Yes, but the mean is also given in brackets. We reported the median for the length and elevation drop, as the mean can be biased to the few outliers.**

**Addressed: L585-L587.**

*L400: Do you consider the same study period of Carrivick and Tweed (2016)? Their study has been published 6 years ago, so the database is shorter.*

**This is correct. However, the large difference stands. We have now clarified that in the text (i.e. over the same period we have 854 vs 300 deaths).**

**Addressed: L602-L605.**

*L423: Did your appraisal account for road disruptions? What is a 'ripple effect'?*

**Unfortunately, there is no appropriate documentation of impacts on roads, but we do mention that this is the biggest concern based on the reports collected in this database (see infrastructure impacts). Together with the change of wording of 'ripple' to 'knock on' effects, a common term when talking about climate risks we have now expanded on this explanation. As we note we also believe that this indirect effect of GLOFs could however be further investigated.**

**Addressed: L629.**

*L426: are thee minimum and maximum values behind the mean?*

**Yes, we have clarified that now.**

**Addressed: L634.**

*L447: It is not expected in ESSD that authors investigate these trends or provide deeper mechanistic insights. This phrase therefore can be deleted (also in L315 and L336).*

**Thanks, we have now removed these statements.**

**Addressed: L662; L460 and L505.**

*L454-456: I could not follow this statement, please revise.*

**Our argument here is that GLOFs in the region have received too much attention if you will. This is based on our experience looking at hazards more holistically. Generally, projects in high mountain and cryosphere hazards are always associated to GLOFs, hence also any large-scale financing of projects is made available when it addresses GLOFs. This has caused local reports to immediately call every mass flow a GLOF, even if no lake whatsoever was involved. The high number of studies is a by product of this focus. Other hazards, like avalanches, debris flows, or other mass movements associated to permafrost change for example have hardly received any attention, even though they often had higher impacts. We have now revised this sentence considerably to make our argument clear and why this is important when critically reflecting on GLOF data.**

**Addressed: L671-L673.**

*L478: The uncertainty and completeness of this variable (and many others) is not assessed or discussed in greater detail. Suggest to extend the associated paragraphs.*

**We agree that this has so far been weakly discussed. We now provide an in-depth discussion of these issues in the section on 'Data structure and variables' (New section 2.2) and pick this up here in the Discussion again.**

**Addressed: L151; L431**

*L496-499: Please add to the discussion how you would achieve that.*

**We have now expanded on this end of the discussion with more detail, proposing a way forward.**

**Addressed: L684-L685.**

L500: The data availability statement usually comes before the conclusions.

**Revised.**

**Addressed: L686.**

L575: Please use the published version of that paper.

**Revised.**

**Addressed: L815.**

**References**

Erman, Alvina, Sophie Anne De Vries Robbe, Stephan Fabian Thies, Kayenat Kabir, and Mirai Maruo. 2021. "Gender Dimensions of Disaster Risk and Resilience." Washington D.C.: World Bank. http://hdl.handle.net/10986/35202.

Glazirin, Gleb. 2010. "A Century of Investigations on Outbursts of the Ice-Dammed Lake Merzbacher (Central Tien Shan)." *Austrian Journal of Earth Sciences* 103: 171–79.

Kääb, Andreas, Myléne Jacquemart, Adrien Gilbert, Silvan Leinss, Luc Girod, Christian Huggel, Daniel Falaschi, et al. 2021. "Sudden Large-Volume Detachments of Low-Angle Mountain Glaciers - More Frequent than Thought." *The Cryosphere* 15: 1751–85. https://doi.org/10.5194/tc-2020-243.

Kingslake, Jonathan, and Felix Ng. 2013. "Quantifying the Predictability of the Timing of Jökulhlaups from Merzbacher Lake, Kyrgyzstan." *Journal of Glaciology* 59 (217): 805–18. https://doi.org/10.3189/2013JoG12J156.

Mankoff, Kenneth D., Xavier Fettweis, Peter L. Langen, Martin Stendel, Kristian K. Kjeldsen, Nanna B. Karlsson, Brice Noël, et al. 2021. "Greenland Ice Sheet Mass Balance from 1840 through next Week." *Earth System Science Data* 13 (10): 5001–25. https://doi.org/10.5194/essd-13-5001-2021.

Medeu, A. R., V. P. Blagoveshchenskiy, T. S. Gulyayeva, and S. U. Ranova. 2019. "Debris Flow Activity in Trans-Ili Alatau in the 20th — Early 21st Centuries." *Geography and Natural Resources* 40 (3): 292–98. https://doi.org/10.1134/S1875372819030120.

Ministry of Planning Development & Special Initiatives. 2022. "Pakistan Floods 2022: Post-Disaster Needs Assessment (PDNA)." Islamabad, Pakistan. https://www.undp.org/pakistan/publications/pakistan-floods-2022-post-disaster-needs-assessment-pdna.

Nie, Yong, Qiao Liu, Jida Wang, Yili Zhang, Yongwei Sheng, and Shiyin Liu. 2018. "An Inventory of Historical Glacial Lake Outburst Floods in the Himalayas Based on Remote Sensing Observations and Geomorphological Analysis." *Geomorphology* 308: 91–106. https://doi.org/10.1016/j.geomorph.2018.02.002.

Uddin, Kabir, and Mir A. Matin. 2021. "Potential Flood Hazard Zonation and Flood Shelter Suitability Mapping for Disaster Risk Mitigation in Bangladesh Using Geospatial Technology." *Progress in Disaster Science* 11: 100185. https://doi.org/10.1016/j.pdisas.2021.100185.

UNESCAP. 2022. "Background Paper for Regional Consultation on Facilitating Innovative Action on Disability-Inclusive and Gender-Responsive DRR." Bangkok, Thailand: UNESCAP.

Welty, Ethan, Michael Zemp, Francisco Navarro, Matthias Huss, Johannes J. Fürst, Isabelle Gärtner-Roer, Johannes Landmann, et al. 2020. "Worldwide Version-Controlled Database of Glacier Thickness Observations." *Earth System Science Data* 12 (4): 3039–55. https://doi.org/10.5194/essd-12-3039-2020.

Dear Reviewer,

We are very grateful for the close reading of the manuscript, the appreciation for making the data accessible beyond academia and also your concerns regarding the quality of impact data and information through local knowledge. We appreciate that these are issues that need to be taken seriously and respond to them point by point in bold below, with your original review kept in cursive.

*Dear editor and authors*

*In this manuscript, Shrestha et al., presented a comprehensive database of Glacier Lake Outburst Floods (GLOFs) in High Mountain Asia (HMA). The authors combined three databases of glacier lakes in this work. The compilation of the GLOF database is largely based on literature review of articles from different sources including peer-reviewed papers, book chapters, technical reports as well as online news articles. The authors also take local knowledge into consideration, which is believed to be necessary considering the issue with under or over reporting of hazards in rural areas in HMA. However, this also brings challenges to the reliability of the sources. The novelty of this database lies in its inclusion of impacts of these GLOFs downstream, which could be difficult to quantify, fact-check and describe in a single .csv file. The authors have also created an interactive map and dashboard for visualization and quick check for non-academic users. I praise the efforts that have been put on compiling such a database, which has great value in creating vulnerability assessments and hazard adaptation plans for mountain communities.*

*However, I also have a few major comments on how the article is written regarding ESSD guideline and how the data is archived, and quality controlled.*

1) *First, the authors spent a lot of efforts on analyzing and interpreting the data in Sect. 3 and 4. Of course, an overall statistic of the data could be included. However, I believe, since the article is about presenting a dataset, the emphasis should be put on elaborating the methods used to produce the data, the choices of the variables, analyzing the quality, uncertainties, and limitations in the data and how it could be useful in other studies.*

   **We agree that the manuscript so far is quite heavy on going in further depth rather than only pointing out what further potential would have been. This is especially the case for the discussion of lakes (which we have now completely removed). As you point out we think that highlighting the general nature (i.e. Figure 2), statistics (i.e. Figure 3 and 4) as well as potential avenues of investigation (Figure 7/8) is warranted within a data journal, while the analysis vs lake types for example goes too far. We have therefore removed these parts (while the baseline data as well as the connection of the database to lake databases of course remains). We furthermore now add a more critical reflection on data quality, uncertainty (as far as quantifiable) as well as general limitations under section 3.1.**

   **Addressed: L151 and L384.**

2) *The methodology about how the lake dataset and GLOFs dataset were derived are not detailed enough. I understand that the lake data was compiled from three earlier datasets from different years. But it is not clear how they are different in terms of*

*coverage and quality, if the three datasets are merged or used for GLOFs happened in different years separately, and if they are merged what the rule for merging is. For the GLOF dataset, the authors stated briefly that the data was derived by reviewing articles and interviews from different sources and verified by satellite imageries, and that false reported events are removed. But I think more details are need in describing and discussing this process as from this description the readers have no idea how reliable these derived data are. And the removed cases need more vigorous justification. Since this article is about the dataset not the interpretation of the data more discussion could be put in these aspects.*

**Following also on the response to the previous questions, we agree that there needs to be some shift in focus. The first part of the question is solved by removing the section on lakes to the minimum necessary as their source (but not describing the datasets), the second is now addressed by having a more detailed discussion of the process in section 3.1.**

**Addressed: L264 (removed) and L384.**

3) *In terms of the datasets, the authors indicate that the dataset is publicly available on ICIMOD data portal (https://doi.org/10.26066/RDS.1973283). I assume this will be the main platform for downloading the data. However, the HMAGLOFDB_v1.0.csv file downloaded from there is not accompanied by either a metadata file, a description file or the HMAGLOFDB_removed.csv file. Thus, the user who downloads the data from there has no idea what each column in the data file means. I later realize that those files are included on the Github repository. But in the ICIMOD data port there is no mentioning of the Github repository. It would be nice to centralize these different bits of data or at least link them together to be more user friendly, especially for non-academic users since they are considered as important stakeholders in the article.*

**Thank you for making this point. Our decision to have two locations for our database, the RDS as well as Github, was indeed a bit of a challenge that stems from our motivation to make this data as much accessible as possible to both non-academics (RDS, established in the region as a tool that aims to ease the challenge of data sharing between countries) as well as academics (github, making it better traceable and integrateable into future analysis). We understand that this was a bit confusing and now make clear in both repositories the presence of the other. We have now also added the Metadata and the removed file to the RDS. However, we would also like to note that on github the database will be continuously updated in future (i.e. as soon as new events are recorded and checked individually), while on RDS this is only done annually after a revision of the data.**

**Addressed: L687 and L690.**

4) *My biggest concern is the criteria for choosing the mechanism involved in lake breach or drainage. It is not clear to me how the authors decided to adopt a definite mechanism or mark it as unknown. I did not check all the GLOF events in the data files. But for the lakes I checked (next to Kyagar glacier and Khurdopin glacier) there seem to be some issues with this.*
*The 34 GLOFs of the lake next to Kyagar glacier are well documented in different articles, which are cited in the HMAGLOFDB_v1.0.csv file. The GLOFs before 2018 were believed to be triggered by ice-dam failure linked to subglacial drainage. And the one in 2018 was more likely to be linked to overtopping. However, the entry for the mechanism is unknown. I don't know what the authors' reasoning behind that.*

*The GLOF next to Khurdopin glacier are marked as caused by 'englacial tunnels. However, Bazai et al. (2022) used a subglacial hydrology model to simulate the sudden drainage. Clearly, Bazai et al. (2022) thought the drainage was likely to be linked to subglacial drainage system. The authors of this manuscript have cited Bazai et al. (2022) but decided to adopt englacial tunnel as the mechanism.*

**Thanks for noting these challenges. Specifically, for Kyagar as well as Khurdopin, we have made an error in the database – these are definitely subglacial drainage mechanisms at play and we have corrected this error now and have redone Figure 4 and adapted the text. We regret this error. As for the many other cases, we generally do only add a known mechanism when this is either documented from the source publication, the news item or without doubt visible from satellite imagery. In most cases this is simply not given, and while we could make an 'educated guess' we believe this would not be beneficial for further analysis. We hope to gather more evidence in future to potentially complete this part, but as of now are not able to do so considering the limited documentation regarding individual events.**

**Addressed: Corrected in the database file.**

5) *There are many events that only recorded in 'this study'. We have no idea how they are identified and quality controlled.*

**We apologize for the omission in this case. While for some of the events we have provided documentation from other sources (i.e. non-academic, news reports etc) there were indeed many cases where this was missing. As the co-authors are responsible for monitoring mountain hazards in the region, either in the field or applying remote sensing, there are a number of cases we are aware of from our daily work. Some of these have been detailed in technical reports** (e.g. Ashraf et al. 2015)**, which are however not publicly accessible. In all cases where this was still feasible we checked all sites with satellite imagery as we did for other cases, following** (Zheng et al. 2021)**. For some of the cases (e.g. the ones in Afghanistan) co-authors were responsible for the rapid response missions, visiting the field sites including the sources. We have not included cases where local populations reported GLOFs where all evidence points to simply debris flows (i.e. no lake source available).**

*Besides, the major issues I also have a few minor comments:*

*L90: (Chen, et al., 2021) -> Chen et al. (2021). There are also some other places that the citations are written not according to the convention.*

**Thank you, revised throughout the manuscript.**

*L115: This kind of statement or practice does not seem to be very rigorous. The authors have excluded many cases that might have been caused by debris flows but include cases that have happened far away from any glacier just because they appear in a landscape that was most likely glaciated at one point?*

**We agree that this is confusing but argue this is due to our formulation. The events we included are indeed all from glaciated terrain, where ice is still present even if not necessarily part of the current inventories anymore. The Lang Co event is actually the only case where no more inventorized glaciers are close, but very recent glacier cover is likely given the geomorphology and rock glacier presence is likely. For other cases where we report 'no glacier' it is simply not possible from the source or the satellite imagery to define which glacier would**

be the potential source of melt water from the existing inventories. We have reformulated the text in this paragraph as below:

*We also record GLOFs that cannot be directly associated to a glacier, either because from the source or satellite imagery it is not clear which glacier upstream feeds into the lake or because there is no adjacent glacier in any of the available inventories. (L138-L140).*

*L120-124: The information in this part is not included in the dataset but only describes how the data is analyzed for the discussion in Sect.4. Following my major comment no.1 I don't know if this should be put here or be included at all.*

**The aggregation is also important for Figure 3, which we believe is important to show the overall distribution of the data we have compiled, and hence we would prefer to keep it. We have however considerably shorted the discussion in section 4 (L431), following the suggestions above.**

*L193-194: In which place did the other 0.8% happen?*

**In China (28.4%) and Kyrgyzstan (24.7%), we had rounded the numbers. Revised.**

**Addressed: L307-L307.**

*L 339-342: Need some references here.*

**Following the suggestions to considerably shorten the discussion as well as the data on lakes itself, this section has been removed completely.**

**Addressed: L512-L515 (removed).**

*Fig. 2: It is not clear why this figure should be included and why the pictures of these glaciers are selected. These pictures are not a part of the database; or should they be included as a part of the database? Something could be a reference is the Norwegian Water Directorate GLOF map ([http://glacier.nve.no/Glacier/viewer/GLOF/en/](http://glacier.nve.no/Glacier/viewer/GLOF/en/)).*

**The figure is included to visualize the type of GLOFs that are included in this dataset and also visualizes to the reader what these types would look like and how we assess them in the field. All the shown examples are from the database. To include the photos into the database is the plan and will eventually be realized on the dashboard ([https://experience.arcgis.com/experience/20a0ef1d86ec4a77b2744df9e4952 14e](https://experience.arcgis.com/experience/20a0ef1d86ec4a77b2744df9e495214e)) where the photos available of a GLOF will appear as you click on the event. However, the development of this dashboard is lagging behind the publication and is subject to staff availability at our employer and hence could not be brought to the final status of the manuscript.**

*Fig. 3 Maybe it is better to use another colormap to be color-blind friendly.*

**Thanks for the suggestion, the colormap has been changed to a color-blind friendly scale.**

**Addressed: L297(Figure 3).**

*Fig. 4 A Should the cause of lake appearing be 'glacier melt' or 'glacier retreat'? Or it means something else?*

**This is again based on the information from studies. While some of the lakes where also formed of course as ice retreats, the information generally pertains**

**to melt water provision, irrespective of retreat or stability. We have now explained that better in the caption.**

**Addressed: L379-L380 (Figure 4).**

*Fig. 6 Not quite sure what the x-axes represent.*

**We have decided to remove the Figure as it comprises analysis that goes too far in depth.**

**Addressed: L545 (removed).**

---

## Author Response (AR2)

Dear Katrin Lindbäck,

Thank you for the careful consideration of the resubmitted version. We have addressed all points below and list any further small edits.

As for the 'development version' we have followed the approach from (Welty et al., 2020), who use GitLab as their development version, which they then push to Zenodo with new releases. The approach followed here is the same. We agree that the phrasing has been slightly unclear and have now made this clear in the "Data availability" section. We use this as the repository for code and have a separate folder for all GLOF events. The advantage of the development version is that our filling of the database is transparent, as any changes we make to the database on GitHub will be forever traceable (while on Zenodo this is somewhat more difficult to follow).

We still have the visualization operational, and this will be available in future (complemented by other datasets in maps of the same design). The reason we removed it from the paper is our concern that the clumsy link to the ArcGIS map may change in future. The ArcGIS server is owned by our institution, and whether they renew the subscription in years to come is out of our control. We have however added the link to the ReadMe on the database, from where it will be accessible at all times, and for the case that we need to migrate the database in future to another server, the link can be updated (as we have full and permanent control over the database). We hope this is agreeable and would appreciate any other suggestion in this direction to make it compatible with ESSD standards.

We also agree that the RDS structure with login is not ideal and have hence removed it from the manuscript. We have retained the link to this DOI in the database itself where it can be accessed.

Other edits:

Figure 1: The spelling of Dzhungarsky Alatau was incorrect and has now been amended.

Data availability statement: We have updated the DOI to the stable DOI that will in future lead immediately to the most recent version of the database.

We hope these edits satisfy ESSD standards and are otherwise happy to discuss further amendments.

Kind regards,

Finu Shrestha and Jakob Steiner

---

## Author Response (AR3)

Dear Editor,

We express our gratitude for accepting our manuscript for final publication in ESSD. We have carefully reviewed the manuscript and made a few updates to address typos and a few syntax/grammar errors we noticed. Most importantly we have changed 'glacier' to 'glacial' in the title (ln 2) and abstract (ln 18) – this somehow slipped our attention. For GLOFs both terms are used, not very consistently throughout the literature but we wanted to align it to the way we had actually used it throughout the text in the end.

The manuscript has been shared with all the authors for their inspection and has received their approval for the final submission. The revised manuscript has been adhered to the journal's submission guidelines.

We want to assure that the content of the previously shared manuscript remains unchanged in the recently uploaded version. Additionally, we have appropriately credited and acknowledged individuals for their data and support.

Kind regards,

Finu Shrestha, Jakob Steiner and the teams